# Enhanced feature matching in single-cell proteomics characterizes IFN-γ response and co-existence of cell states

Karl K. Krull [1,2,3], Syed Azmal Ali [1] & Jeroen Krijgsveld [1,3] ✉

Proteome analysis by data-independent acquisition (DIA) has become a powerful approach to obtain deep proteome coverage, and has gained recent traction for label-free analysis of single cells. However, optimal experimental design for DIA-based single-cell proteomics has not been fully explored, and performance metrics of subsequent data analysis tools remain to be evaluated. Therefore, we here formalize and comprehensively evaluate a DIA data analysis strategy that exploits the co-analysis of low-input samples with a so-called matching enhancer (ME) of higher input, to increase sensitivity, proteome coverage, and data completeness. We assess the matching specificity of DIA-ME by a two-proteome model, and demonstrate that false discovery and false transfer are maintained at low levels when using DIA-NN software, while preserving quantification accuracy. We apply DIA-ME to investigate the proteome response of U-2 OS cells to interferon gamma (IFN-γ) in single cells, and recapitulate the time-resolved induction of IFN-γ response proteins as observed in bulk material. Moreover, we uncover co- and anti-correlating patterns of protein expression within the same cell, indicating mutually exclusive protein modules and the co-existence of different cell states. Collectively our data show that DIA-ME is a powerful, scalable, and easy-to-implement strategy for single-cell proteomics.

The increased awareness of biological heterogeneity within cell populations, evidenced by profound differences in gene transcription among individual cells[1–4] has changed our understanding of the confines of what is regarded as the same cell type[5,6]. This phenomenon helped to explain biological transitions as a continuous process rather than a collection of discrete steps and could also have important implications for understanding and treating diseases[7]. However, a description only at the transcriptomic level lacks information about post-transcriptional regulation that translate into functional proteomic changes[8–10], therefore requiring techniques for the proteomic investigation of individual cells. While single-cell RNA sequencing has become routinely accessible, single-cell proteomics (SCP) has

undergone a more recent but steep development, benefitting from increased sensitivity offered by novel sample preparation workflows and mass spectrometric instrumentation[10–12].

Yet, SCP still suffers from several limitations in proteomic depth and throughput. Since the material from individual cells is scarce, several studies focused on reducing losses during sample preparation by process miniaturization[13,14] and avoiding surface adsorption[15,16]. As one important example, the SCoPE-MS (single-cell proteomics by mass spectrometry) methodology uses sample multiplexing via TMT-labeling to concomitantly increase analyte concentration and throughput[17,18]. However, recent reports propose that the inclusion of carrier channels in this method can

[1]German Cancer Research Center (DKFZ), Heidelberg, Division of Proteomics of Stem Cells and Cancer, Heidelberg, Germany. [2]Heidelberg University, Faculty of Biosciences, Heidelberg, Germany. [3]Heidelberg University, Medical Faculty, Heidelberg, Germany. ✉e-mail: j.krijgsveld@dkfz.de

compress reporter ion ratios, jeopardizing the accuracy of protein quantification[19–21]. For this reason, label-free approaches have become a popular alternative, where in particular the use of collective precursor isolation in MS via data-independent acquisition (DIA) showed enhanced sampling depth in shorter time compared to data-dependent acquisition (DDA)[22]. Nevertheless, DIA produces chimeric MS/MS spectra containing highly convoluted mixtures of simultaneously fragmented precursors and their fragment ions, thus demanding special ways of data analysis. Here, the introduction of diaPASEF was pivotal, enhancing the signal-to-noise ratio by excluding singly charged ions before MS/MS acquisition[23]. Still, to cope with the complexity of the resulting spectra, pre-generated libraries comprising unambiguous information on peptides and their fragments remained inevitable[23,24]. Particularly, this poses challenges in SCP as library-based analyses require time-intensive data collection from preceding runs, varying with instruments and projects, and they demand higher sample input even if cell types are scarce. Novel software tools have partly overcome these constraints by direct (i.e. library-free) analysis of DIA spectra, of which Spectronaut[25] and DIA-NN[26] are the most widely used. Although these tools handle data in slightly different manners, e.g. either utilizing a peptide-centric (DIA-NN) or spectrum-centric approach (Spectronaut), both consist of a two-step process, in which the initial assignment is followed by storing the information in an internal library that is subsequently used to re-analyze the data[26–29]. This procedure shares a conceptual resemblance with the match-between-runs (MBR) algorithm that is commonly used in DDA approaches, as it aims to recover unidentified features to mitigate missing values across experiments. Since high data completeness is crucial when analyzing large sample cohorts, as is usually the case in single-cell studies, MBR has frequently been applied in DIA workflows[14,30–36]. In some of these studies, it has been empirically shown that inclusion of higher input samples conveniently leads to better data coverage, however, the performance characteristics and optimal experimental conditions have not been formally addressed. Therefore, we here performed a systematic evaluation of such an approach, with the specific aim to use its huge potential to establish an optimized workflow for label-free proteomics of single-cells. The core concept of the approach is that low-input samples of interest are co-analyzed with high-input samples, that we here term matching enhancers (MEs), and that serve to increase sensitivity by identifying equivalent features during the matching step in DIA data analysis. Hence, we hereby call this method DIA-ME as an experimental strategy that improves proteome coverage and data completeness in low-input DIA data. In this study, we assessed the quality of matching in DIA-ME, and determined if diminished signal intensity in low-input data leads to false feature matching or skewed quantitative precision. In addition, by employing a benchmark dataset of a two-species model system, we evaluated how quality of matching depends on the peptide amount used in the donor data set and how FDR-control can be preserved in this process. At the same time, we assessed differences in performance of DIA-ME when using Spectronaut and DIA-NN as data processing suites, and provide guidelines how data quality can be maintained when using these tools. While spectral libraries are frequently used in SCP to achieve higher sensitivity[14,34–38], we show that DIA-ME effectively circumvents this requirement by considerably increasing proteomic depth in library-free searches. Using injection amounts as small as that of a single cell (200 pg), we demonstrate that DIA-ME improves proteome coverage with high quantitative accuracy compared to conventional data analysis, resulting in the characterization of the IFN-γ-responsive proteome that was highly similar to that obtained by a bulk analysis. Finally, we highlight the DIA-ME-assisted analysis of 143 U-2 OS cells, which revealed oppositely regulated proteome programs, pointing to mutually exclusive protein expression and the co-existence of different cell states within the same population.

## Results

### The principle of enhanced matching in DIA-ME

A key principle to achieve high data completeness in DIA data is the transfer of peptide information across multiple samples in the same set. This functionality is at the core of Spectronaut[25] and DIA-NN[26], and we endeavored to further exploit its capabilities in low-input proteomic data, which usually suffers from missing values and limited sensitivity. Specifically, both tools handle DIA data in a 2-step procedure (despite fundamental differences in their architecture), storing information on precursor and fragment ions in an internal spectral library after the initial identification step, which is then used in a second pass to specifically extract elution profiles from other runs that are analyzed in parallel. This last step considerably reduces missing values across samples, and in the following we refer to it as "MBR" (match-between-runs), although this has been originally coined for DDA data. We speculated that expanding the resources of the internal library could help to concomitantly improve sensitivity and data completeness of proteomic experiments, especially when data are sparse as in single-cell analyses. To this end, we here formalize and evaluate this concept, wherein files with low-input runs are jointly analyzed with runs from higher sample input, effectively serving as a reference database. We termed the files from higher input samples "matching enhancers" (MEs), as they contribute to the creation of an enlarged internal library that is used to extract similar signals from low-input runs to improve sensitivity of the analysis (Fig. 1A), and we refer to the overall approach as DIA-ME. DIA-ME is easy to implement in existing proteomic workflows, as it only requires running a few MEs together with a series of low-input samples, followed by analyzing the collective data in existing software tools. In this work, we aimed to critically assess the performance of DIA-ME, and apply it to the investigation of single cells.

### DIA-ME expands proteomic coverage from low-input data

Having conceptualized the DIA-ME workflow, we aimed to assess the gain in proteome coverage that can be achieved, to determine the optimal size of the ME samples, and to evaluate potential false matching events between MEs and low-input samples. To benchmark this, we generated a ME sample set that consists of mixed HeLa and *Escherichia coli* (*E.coli*) peptides, where the former serve as a "donor" for feature matching to a low-input human "acceptor" sample, while the latter introduce features that are used to assess false transfers. Specifically, we prepared samples of HeLa digests spiked with different ratios of *E.coli* peptides (5–20%), and adding up to different total peptide amounts of 1–100 ng (Fig. 1B). We used an active LC gradient of 15 min for LC-MS analysis, and acquired data on a Bruker timsTOF Pro instrument in diaPASEF mode (Supplementary Fig. 1A). Analysis of ME samples in DIA-NN led to the identification of approx. 8000 and 3500 protein groups in 100-ng and 1-ng samples, respectively (Supplementary Fig. 1B). As we intended, *E.coli* peptide intensities increased by more than two orders of magnitude, thereby scaling with the amount of spiked *E.coli* peptides (Supplementary Fig. 1C, D). Consequently, our two-proteome samples contained highly variable records of *E.coli* features ranging from around 500 to 6000 identified peptides, making it a suitable system to evaluate correct matching in low-input data.

Next, we analyzed a set of 1-ng (non-spiked) HeLa peptide samples under the same LC-MS conditions (Supplementary Fig. 2) and investigated the effect of different processing methods (Fig. 1B) (Supplementary Data 1). Conventional library-free analysis via DIA-NN identified on average 2800 proteins, which was increased to 3300 when applying MBR (2600 and 3100 proteins, respectively, when using Spectronaut), confirming the benefit of activating this function (Fig. 1C). Remarkably, co-analysis of HeLa with ME samples drastically improved proteome coverage even further, reaching up to approx. 4650 proteins in 1-ng HeLa samples when analyzed along with 10-ng MEs (i.e. 10× ME) by either DIA-NN or Spectronaut (Fig. 1C). This improvement corresponds to an increase of around 60% and 70% in

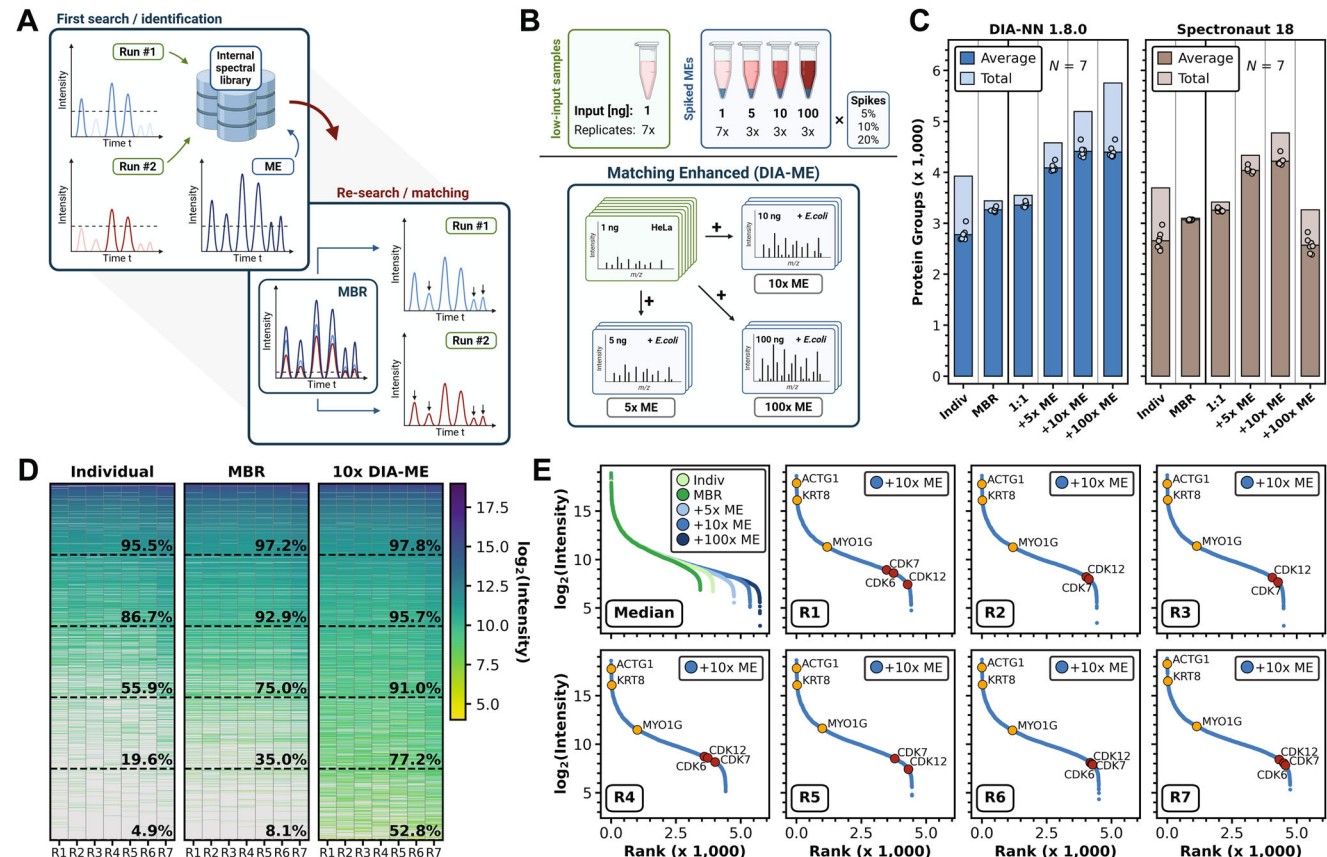

**Fig. 1 | DIA-ME enables ultra-high sensitivity and data completeness. A** DIA-ME principle: prevalent DIA data analysis is based on a two-step process. Peptides are identified and stored in an internal library, before their information is used to re-analyze (match) runs that are searched in parallel. Providing a high-input sample (Matching enhancer: ME) to the first search, more information can be gathered in the library, which aids to identify low-abundance signals in the low-input runs during matching. **B** Mixed-species experiment to evaluate DIA-ME data analysis (top). Two types of samples: seven low-input replicates containing *H.sapiens* proteome (HeLa, green) and twelve sets of *H.sapiens* samples spiked with *E.coli* K12 proteome (blue). Proteomic mixtures differed in their spiking ratio (5−20%) and total peptide amount (1−100 ng). Resulting files of spiked samples were used to evaluate the DIA-ME data analysis (bottom): low-input *H.sapiens* samples (green) were analyzed with a triplicate of spiked 5-ng (5× ME), 10-ng (10× ME) or 100-ng

(100× ME) runs (blue). **C** Average and total (light) protein groups in the seven non-spiked 1-ng replicates by DIA-NN (blue) and Spectronaut (brown) using different analysis strategies (see **B**). Indiv: individual raw file analysis. MBR: Collective raw files analysis with activated matching. 1:1: Co-analysis with seven spiked 1-ng replicates. Results only shown for analyses involving a spiking ratio of 10%. **D** Heatmap of ranked protein group (PG) intensities for individual, MBR and 10× DIA-ME analysis of 1-ng HeLa replicates (R1 − 7). Six bins (divided by dashed lines) indicate the obtained data completeness in the respective intensity segment per analysis. **E** Upper left: ranked median protein intensities in different data analyses. Others: ranked protein intensities in 1-ng replicates (R1 − 7) after 10× DIA-ME analysis. Three high- and medium-abundance cytoskeleton proteins (yellow) and three low-abundance cell cycle-related proteins (red) shown, the latter only identified in DIA-ME analysis. Source data are provided as a Source Data file.

comparison to the initially performed individual analysis in DIA-NN and Spectronaut, respectively, and was primarily caused by the re-extraction of features that were additionally identified from ME files (Supplementary Fig. 3). We ranked protein groups based on their reported intensities and demonstrated that additional ME-derived identifications also considerably contributed to an improved data completeness in DIA-ME analysis (Fig. 1D). Dividing the abundance range into equally sized bins, we further revealed that newly identified proteins were found across the entire scale, but mainly helped to reduce missing values in the low intensity area, thereby increasing data completeness in the two lowest bins from 35.0% to 77.2% and from 8.1% to 52.8%, respectively (in comparison to MBR alone). Moreover, DIA-ME-enabled identification of low-abundance proteins led to an expansion of the protein dynamic range by a full order of magnitude (Fig. 1E). This observation was accompanied by the consistent quantification of several low-abundance cell cycle-dependent proteins that were not identified using MBR alone. Notably, we recognized that the average number of identified proteins saturated for the DIA-ME analysis at 10-ng (i.e. 10× MEs) in DIA-NN (Fig. 1C), which suggests that identifications cannot be increased infinitely, and that identity transfer

does not occur randomly (explored in more detail below). Analysis of the same data by Spectronaut showed very similar trends, although overall identifying fewer proteins (Fig. 1C). Here, the most notable observation was the drop in protein identifications when using 100× DIA-ME. We did not further investigate this phenomenon, but assume that feature matching might be impeded in this software due to the large differences in signal intensities. Collectively, our results show that the concept of extensive feature matching in DIA-ME benefits low-input data by decreasing missing values, augmenting sensitivity, and resulting in enhanced proteome coverage.

## DIA-ME improves qualitative reliability in low-input data

Encouraged by these results, we aimed to assess the reliability of protein identification and feature matching in DIA-ME. Since the vast majority of *E. coli* peptides are not shared with the human proteome, we evaluated coverage of species-specific proteomes in non-spiked 1-ng samples before and after they were co-analyzed with *E. coli*-containing ME samples. In this way, we used the *E. coli* peptides in ME samples as a matching resource to estimate erroneous feature assignment (Supplementary Data 2 and 3).

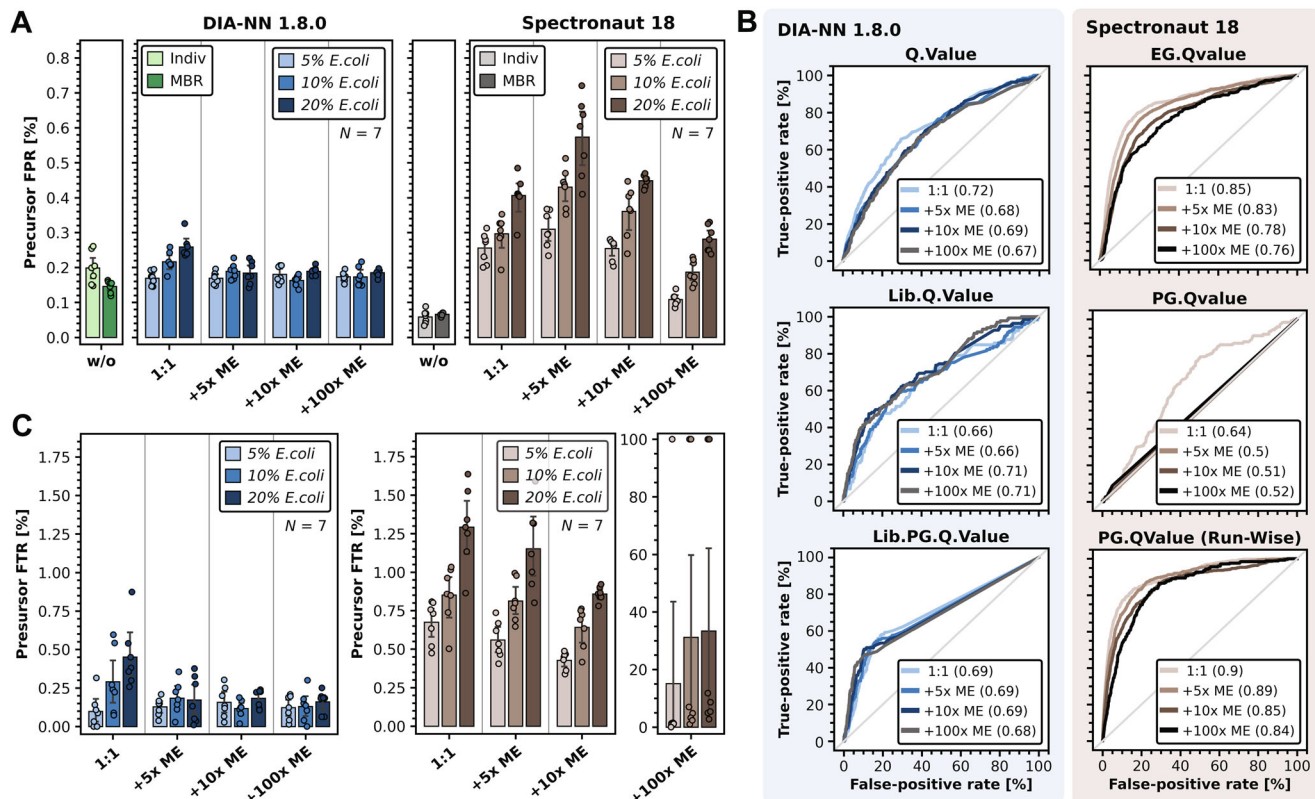

**Fig. 2 | Low FPR and reliable feature matching in DIA-ME. A** False positive rate, i.e. percentage of detected *E.coli* peptides, in non-spiked 1-ng *H.sapiens* samples (*N* = 7) for different types of data analysis and DIA software. Analyses without spiked samples, i.e. without entrapped matching, are indicated in green and grey (light-: without MBR, dark-: MBR), while co-analyses with spiked samples are indicated in blue and brown for DIA-NN and Spectronaut, respectively. The shade of the color indicates the *E.coli* spiking ratio. Error bars are shown as mean ± sd. **B** Receiver operating characteristics (ROC) of default q-value filters in DIA-NN (left) and Spectronaut (right) for data analyses involving spiked ME samples with 10%

spiking ratio (DIA-NN: light blue (1:1) to dark grey (100× DIA-ME), Spectronaut: light brown (1:1) to black (100× DIA-ME)). Areas under ROC (AUROC) are indicated in parentheses, while the diagonal line represents a random classification. **C** False transfer rate, i.e. percentage of *E.coli* peptides among identifications that were transferred by matching, in non-spiked 1-ng *H.sapiens* samples (*N* = 7) for different types of data analysis and DIA software. Rate was set to 100% when fewer *H.sapiens* peptides but more *E.coli* peptides were identified after matching. Color-coding as in (**A**). Error bars are shown as mean ± sd. Source data are provided as a Source Data file.

We first assessed species-specific identifications in the previous analyses and determined the collective false-positive rate (FPR) of *E.coli* peptides (Fig. 2A). Using an entrapment approach by combining a human and *E.coli* fasta file, we found reasonably low FPRs of 0.2% and 0.06% in DIA-NN and Spectronaut, respectively, for the library-free analysis of individual raw files. While samples in this analysis did not contain *E.coli* peptides, the number of false positives increased when they were co-analyzed with spiked samples. In the experiment of equivalent input amounts (i.e. 1:1), this led to the FPR reaching up to 0.57% in Spectronaut (Fig. 2A), which corresponded to around 1.4% false positives on protein level (Supplementary Fig. 4A). This observation has important practical implication as it represents the error susceptibility of matching among multiple low-input samples, which is standard in most studies. Furthermore, Spectronaut's ability to correctly assign features was dependent on the data provided for matching, since the FPR scaled with the spiking ratio (Fig. 2A). In practice, this might lead to problems if the proteome composition differs substantially between experimental conditions, e.g. when comparing different cell types. In contrast, DIA-NN displayed consistently low levels of false-positive identifications on peptide and protein level (Fig. 2A and Supplementary Fig. 4A). Remarkably, the application of DIA-ME did not lead to an increased FPR and showed a maximum of 0.19%, irrespective of the ME ratio (Fig. 2A). Especially in comparison to individual analysis of raw files (restricted to first pass), these observations suggest that the underlying matching process (second pass) is considerably less controlled in Spectronaut than in

DIA-NN. Moreover, it underscores that joint analysis of low-input and higher peptide amount samples does not compromise protein identification quality in the context of DIA-NN. Here, default filtering sufficed to obtain protein-level FPR below 1% using DIA-ME (Supplementary Fig. 4A), while Spectronaut's FDR control was affected upon the presence of ME samples, and we therefore reinforced the run-specific protein q-value cutoff from 5% to 1% to account for the biased global q-value (PG.Qvalue) (Fig. 2B). However, we still observed more than 1% false positives for the application of DIA-ME, which might be further addressed by changing the filter of the peptide-level PEP score in this software (Supplementary Fig. 5). Interestingly, we did not observe a similar effect on the respective global q-values (Lib.Q.Value & Lib.PG.Q.Value) in DIA-NN (Fig. 2B), however, a moderate cutoff for the run-specific protein q-value (PG.Q.Value; not default) might further reduce false-identifications without sacrificing true positives (sensitivity) (Supplementary Fig. 4B). To estimate the resulting rate of erroneous feature matching in DIA-ME, we assessed the proportion of *E.coli* peptides among newly identified peptides in HeLa samples after matching (false-transfer rate (FTR)) (Fig. 2C). Using Spectronaut, we found elevated FTR, incorrectly transferring between 0.7 and 1.3% of peptides in conventional co-analysis of equal input amounts (i.e. 1:1) (Fig. 2C), which translated to up to 9% of transferred protein identifications (Supplementary Fig. 4B). In comparison, DIA-NN displayed consistently moderate FTRs of 0.1–0.45% in 1:1 and only up to 0.2% across DIA-ME analyses (Fig. 2C). The observation that this is the case even when matching the 1-ng HeLa sample to 100-ng ME samples that

contain 20% *E.coli* peptides indicates high specificity in the matching process that is resilient to the presence of highly abundant interfering peptides. Contrarily, the majority of transferred peptides falsely originated from the *E.coli* proteome for 100× DIA-ME in Spectronaut (Fig. 2C), even though the total number of matched features were minor in this analysis (Fig. 1C). Hence, the addition of MEs in Spectronaut needs to be carefully limited to 10× DIA-ME to achieve optimal proteome coverage and high matching confidence. Interestingly, however, we observed an apparent decline in FTR when employing up to 10× DIA-ME in both software tools (Fig. 2C). Reduced FTR implies that the better spectral quality in MEs potentially facilitates matching in low-input data when compared to regular MBR among equal inputs. Hence, we not only established that DIA-ME leads to an expanded proteome coverage, but also enhances the fidelity of feature matching in low-input data when co-analyzed with MEs.

## Precise and accurate quantification with DIA-ME

After successfully showing confident protein identification driven by DIA-ME, we next probed the quality of protein quantification using this concept in DIA-NN. Since normalization is crucial for reliable

quantification by compensating for differences between injected peptide amounts, we removed rows from the DIA-NN report that contained peptides identified in ME samples before re-normalizing our datasets. Next, we examined different ways of data normalization for low-input DIA data, including MaxLFQ from the *R* package of DIA-NN[26], *iq* normalization[39] and the recently published directLFQ package in Python[40]. Remarkably, utilizing the output of DIA-NN resulted in insufficient quantitative results for low-input data, while directLFQ showed the highest accuracy among the tested normalization strategies (Supplementary Fig. 6A) and it was therefore selected for the following analysis.

To evaluate the resulting protein quantities, we calculated the coefficient of variation (CV) of protein groups as a measure of the variability within different experiments (Fig. 3A). Using directLFQ normalization, we found similar distributions in conventional MBR and DIA-ME analysis with median CV values between 0.16 and 0.19. The slightly higher CVs in DIA-ME experiments may be attributed to the additionally identified proteins, which are generally of lower intensity and are therefore more difficult to quantify precisely (Fig. 3B). However, reassuringly, CVs of newly identified proteins

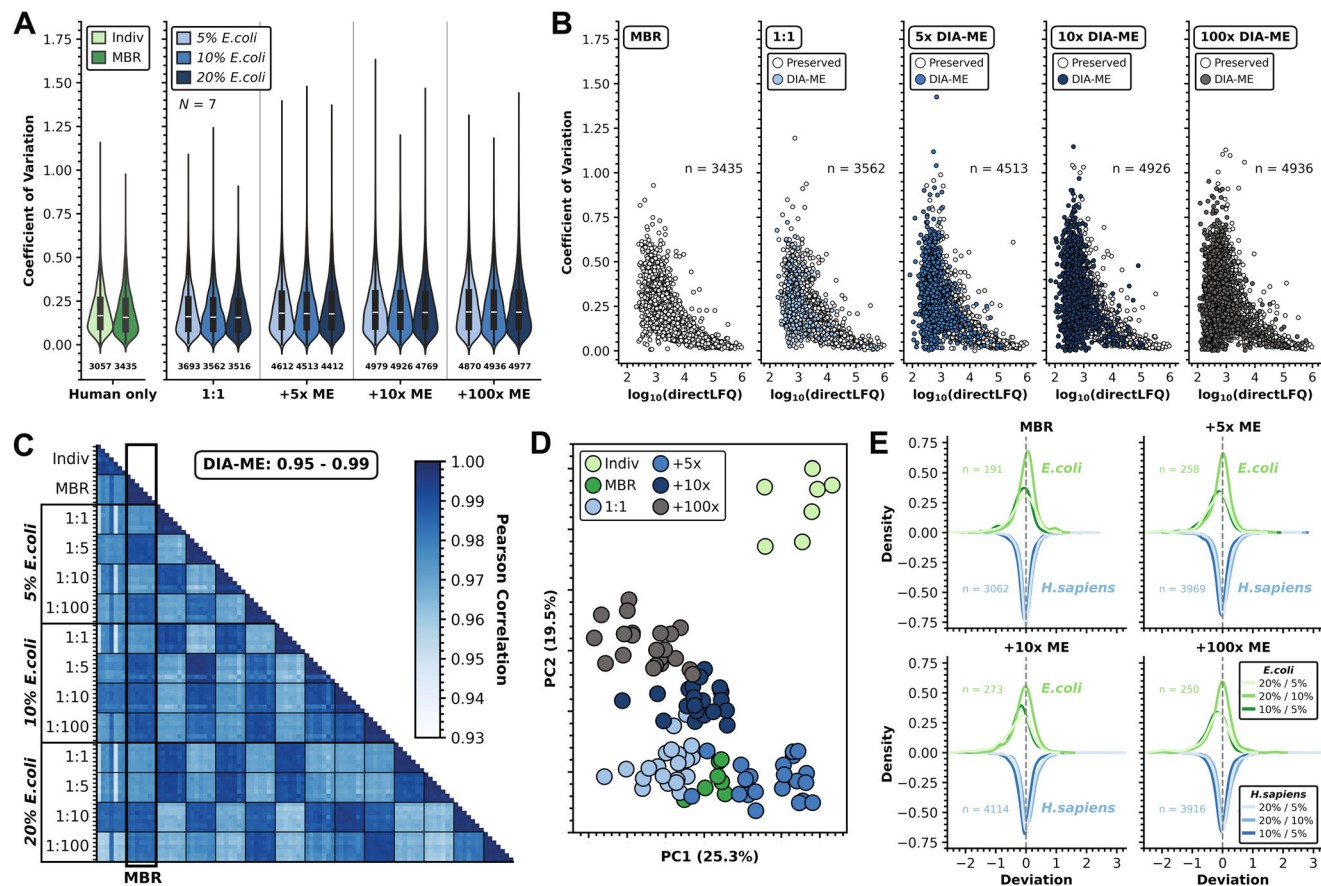

**Fig. 3 | Stable precision and accuracy in protein quantification using directLFQ normalization. A** Violin plot of human protein CVs in 1-ng samples (*N* = 7) with (dark green) and without MBR (light green) on the left and after matching against 1-ng (1:1), 5-ng (5× ME), 10-ng (10× ME) and 100-ng (100× ME) samples on the right. Violins are colored in blue shades according to the spiking ratio in ME samples. Black boxes in the violins show the dispersion of values between the first and third quartile with the white line representing the median of the dataset. Whiskers and violins show the entire range from the minimum to the maximum data point. Number of proteins in each analysis indicated beneath violins. **B** Scatter plot of human protein CV values across 1-ng replicates (*N* = 7) dependent on their reported abundance for different types of data analysis (only shown for 10% spiking). White: proteins already identified in MBR analysis; blues and dark grey: proteins

exclusively found using DIA-ME. Number n of proteins per analysis indicated. **C** Pearson correlation heatmap within and among different types of data analysis. Correlations to conventional MBR analysis are framed. Range of observed correlations in DIA-ME analyses highlighted on the top. **D** Principal component analysis localizing individual replicates from all performed searches in a two-dimensional coordinate system. Data analysis with and without MBR (light-) colored in green, and DIA-ME analyses illustrated in blue shades and dark grey. **E** Density plot showing the deviation from expected ratios in MBR and DIA-ME analyses for *H.sapiens* (blues, inverted density) and low-abundance *E.coli* proteins (greens). Total number n of proteins with at least one ratio in the respective analysis indicated. Source data are provided as a Source Data file.

scaled very similar with their abundance when compared to proteins that were already identified by conventional MBR, testifying to equivalent consistency in the quantification process. As anticipated, Pearson correlations of protein quantities across all previously performed experiments showed enhanced quantitative correlation with activated matching, surpassing individual analyses (Fig. 3C). Interestingly, the application of DIA-ME with varying ME input amounts did not alter this observation and consistently displayed outstanding replicate correlations ranging from 0.95 to 0.99. Moreover, DIA-ME results exhibited strong inter-correlations and aligned with MBR results (indicated by the framed column in Fig. 3C). This finding is further substantiated by the results of the principal component analysis (PCA), revealing cohesive clustering of replicates from the same experiment and with results from MBR analysis (Fig. 3D). Low quantitative variation and remarkably strong correlation both suggest that highly precise protein quantification is maintained in DIA-ME.

To investigate whether DIA-ME affects quantitative accuracy, we assessed if *E.coli* spiking ratios were reflected by relative protein quantification within our spiked 1-ng samples (Supplementary Data 4). Using maxLFQ intensities from DIA-NN, we noticed that protein quantities were estimated inaccurately already when analyzed by conventional MBR (Supplementary Fig. 6A), while protein intensities increased uniformly when samples were co-analyzed with higher input ME samples (Supplementary Fig. 6B). Once we used raw peptide intensities and performed protein quantification in the DIA-NN *R* package, this effect was reversed, indicating that it is caused by the internal peptide normalization in DIA-NN, however, did not lead to accurate quantification (Supplementary Fig. 6A). Meanwhile, re-normalizing peptides by directLFQ solved both issues and resulted in relative *E.coli* quantities that aligned with their expected ratios after MBR analysis (Supplementary Fig. 6A, B). Remarkably, using this normalization, we found that median accuracy was unaffected even for the quantification of +34% human and +43% *E.coli* newly identified proteins after 10x DIA-ME analysis (Fig. 3E), testifying the validity of this method. Not surprisingly, however, we also noticed an increased dispersion around the expected value, resulting from the low abundance of additional identifications. Hence, our findings emphasize the importance of normalization for the analysis of low-input DIA data in general, but also for the co-analysis of samples of different input amounts. Consequently, we recommend removing peptides that were identified in ME samples from the DIA-NN report before performing directLFQ normalization as a downstream procedure after DIA-ME analysis (workflow illustrated in Supplementary Fig. 6C). Collectively, the quantitative analysis of the DIA-ME approach has verified that precision and accuracy can be maintained in the presence of higher input samples, while obtaining significant improvement in proteome coverage.

## DIA-ME enables in-depth analysis of single-cell-like input U-2 OS cells upon IFN-γ treatment

Having thoroughly tested the performance of DIA-ME in defined benchmark samples, we next aimed to evaluate its benefits for low-input quantities that are equivalent to single cells, and in particular, to investigate if DIA-ME improves the ability to detect proteome differences between such samples. To this end, we evaluated if DIA-ME can recapitulate the effect in human osteosarcoma cells (U-2 OS) that were treated with Interferon gamma (IFN-γ), an extensively characterized model to study cellular immune response via JAK/STAT signaling (reviewed in refs. 41–43).

As a starting point, we first generated a reference dataset in a conventional bulk proteomics analysis using 200-ng peptide injections from U-2 OS cells at four time-points during IFN-γ treatment (Supplementary Data 5). Among more than 5,000 identified proteins (Supplementary Fig. 7A) that were quantified with high

reproducibility (Supplementary Fig. 7B), we found several proteins to be differentially regulated at distinct time-points (Supplementary Fig. 7C). While the overall effect of IFN-γ on U-2 OS cells was modest, several proteins indicated strong responses to the treatment with the most pronounced effect on the proteome after 18 h (Supplementary Fig. 7B) and clear distinction between treated and control samples in PCA analysis (Supplementary Fig. 7D). Cluster analysis revealed multiple known IFN-γ-responding proteins, showing gradual and strong upregulation over 24 hours of treatment (Supplementary Fig. 7E). For instance, this cluster contained PMSB8, PSMB9 and PSMB10, the three distinguishing members of the IFN-γ-induced immunoproteasome[44–46], as well as TAP1/2 and TAPBP that translocate peptides produced by the immunoproteasome to the endoplasmic reticulum (ER) for loading onto the nascent MHC class I receptor (B2M and HLAs) (Supplementary Fig. 7F). Moreover, the immunological reaction was reflected by rapid induction of JAK1 and STAT1 (but not STAT2) (Supplementary Fig. 7G), while other pathways, such as mTOR-mediated signaling, are usually activated at a later stage of an interferon treatment[42]. Consequently, over-representation analysis of upregulated proteins after 24 h showed a clear enrichment of immunologically related processes, including response to IFN-γ (Supplementary Fig. 7H). Hence, these data show that the mild proteome response of U-2 OS cells to IFN-γ is detectable in a bulk experiment, identifying known players of the antigen processing and presentation pathway.

We next repeated the IFN-γ treatment experiment, however now using the equivalent of single-cell inputs in a DIA-ME approach. To this end, we collected cells at six time-points during IFN-γ treatment over the course of 24 h from three independent cell cultures (Fig. 4A), and used triplicate injections of 200 pg per sample, the estimated protein amount of a single U-2 OS cell. For matching purposes, we moreover performed single-shot injections of all six time-points and cultures with peptide input amounts of 1, 2 and 10 ng (i.e. 5×, 10×, and 50× MEs), respectively, which we then co-analyzed with our 200-pg samples using DIA-NN.

We used the two extreme time-points (0 h + 24 h) as matching resource for the following analysis (Supplementary Data 6), and increased the average number of identified proteins per 200-pg sample from 1,872 in conventional MBR to 2,594 using DIA-ME (+39%) (Fig. 4B), while using other ME time-point samples did not further improve this result (Supplementary Fig. 8A). Interestingly, this was achieved with 10x ME samples and did not further increase with 50x MEs. Since we similarly observed saturation with the 10x ME samples in the previous experiment (Fig. 1C), we conclude that the optimal amount of MEs is ratio-dependent rather than being determined by the absolute protein amount in the MEs. Analysis of 200-pg samples with 10x DIA-ME comprised a total of approx. 4,200 protein identifications (Fig. 4B), among which 75% were found in at least one replicate per time-point (Supplementary Fig. 8B)), indicating improved proteome coverage and data completeness. The resulting dataset contained more than 99% of the identifications that were also found in MBR, while MBR covered only 53% of proteins identified by DIA-ME (Fig. 4C), effectively leading to an expansion of the dynamic range (Fig. 4D). Crucially, DIA-ME also dramatically increased the number of peptides per protein (Fig. 4E, Supplementary Fig. 8C and D), providing a stronger basis for protein quantification. Indeed, the DIA-ME dataset displayed great quantitative alignment, exhibiting Pearson correlations between 0.88 and 0.99 across time-point and biological replicates (Fig. 4F), while detecting slightly better correlation among earlier and later time-points. This observation is also reflected by the identification of two distinct clusters of earlier and later time-points by PCA analysis (Fig. 4G), indicating changes in the proteome composition over the course of the experiment. Taken together, these data show that DIA-ME provides enhanced proteome coverage in single cell-like (200 pg) samples, benefiting from ME

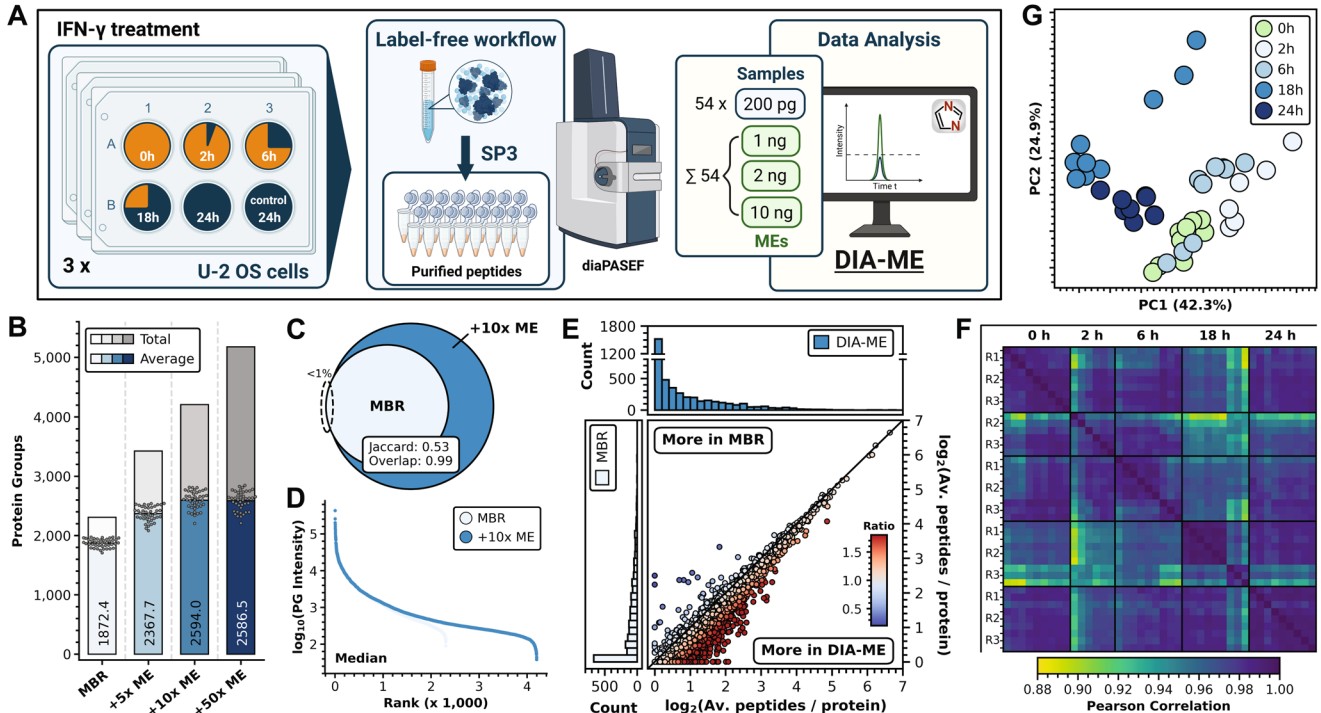

**Fig. 4 | DIA-ME improves proteome coverage in IFN-γ treated U-2 OS cells.**
**A** Experimental scheme of IFN-γ treatment and bulk preparation of U-2 OS cells in the DIA-ME workflow. Six time-points from three biological replicates were collected and samples were diluted to the indicated injection amounts. Three 200-pg injections (blue) and single-injections of 1–10 ng (ME samples, green) per time-point and biological replicate resulted in 108 runs. Obtained files were co-analyzed by DIA-ME in DIA-NN. **B** Total (greys) and average protein groups (blues) identified in 200-pg samples after co-analysis with 1-ng,[5×] 2-ng[10×] and 10-ng[50×] MEs compared to MBR analysis without references samples. Individual protein identifications per sample indicated as grey dots. MEs were derived from control samples before (0 hours) and after 24 h treatment. **C** Venn diagram of identified protein groups in MBR (light blue) and 10× DIA-ME analysis (blue). Overlap (Szymkiewicz–Simpson)

and Jaccard index are given as a measure of the similarity of both populations (see methods). **D** Ranked median protein group intensities for MBR (white) and DIA-ME analysis (blue). **E** Joint plot (center) of histogram distributions (top and left) of average peptides per protein identified in MBR (y-axis, light blue) and DIA-ME analysis (x-axis, blue). The scatter represents proteins identified in both analyses with their color showing the ratio of identified peptide numbers, indicating higher identifications in MBR (blue) or in DIA-ME (red). Peptide number equality (ratio of 1) shown as black line. **F** Pearson correlation heatmap of time-point samples (R1–R3: biological origin), showing correlations from yellow (low) to dark blue (high). **G** Principal component analysis of time-point samples after 10× DIA-ME analysis. Control samples (0 h) shown in light green and treated samples shown in blue shades. Source data are provided as a Source Data file.

samples that are as small as two nanogram, while exhibiting highly reproducible quantification.

To reveal the specific effect of IFN-γ and compare the findings to our reference dataset (Supplementary Fig. 7), we investigated differential protein expression at a depth of more than 3,000 proteins per time-point and detected significant up-regulation of multiple proteins after 24 h (Fig. 5A). As in the bulk (200-ng) samples (Supplementary Fig. 7C), this comprised STAT1, showing the induction of JAK/STAT-mediated signal transduction, and the MHC class I molecules HLA-A/B/C and B2M. Remarkably, the consistent quantification in DIA-ME analysis resulted in similar p-values compared to MBR results (Supplementary Fig. 8E), while it enabled the quantification of 33.5% proteins of the total dataset and of 22.9% of known IFN-γ responders (increase of around 50% and 30%, respectively) (Fig. 5A, C). In this way, DIA-ME allowed the additional identification of low-abundance proteins with high significance, thereby covering the induction of TAP2 and mTOR (Fig. 5A). Notably, the proteins whose expression could now be quantified over time (Fig. 5D, E), showed great similarity with our findings from 200-ng sample analysis (Supplementary Fig. 7F). For instance, we observed gradual upregulation of TAP1/2, TABPB and STAT1, which also grouped together with HLA-A and –C in a hierarchical cluster analysis (Fig. 5B). This cluster also contained calreticulin (CALR), a chaperone described to mediate the peptide loading process of TABP to the MHC class I complex[47] and the transmembrane protein CD40, which acts a mediator in the interaction of antigen-presenting cells with various immune cells[48]. Furthermore, one of the strongest

upregulated proteins was WARS1 (Fig. 5A), a tryptophanyl-tRNA synthetase that is known as a target of IFN-γ signaling[49,50]. We also recognized changes in the proteasome and found, as in the bulk data, the apparent restructuring of the 20S core particle by down-regulation of PSMB5 to PSMB7, and concomitant upregulation of their alternative subunits PMSB8 to PSMB10 that characterize the immunoproteasome[44–46] (Supplementary Fig. 8F). Accordingly, over-representation analysis of upregulated proteins revealed the significant enrichment of several immuniologically relevant processes, including the signaling of IL6/JAK/STAT3, DNA repair and the underlying response to IFN-γ (Fig. 5F). Strikingly, DIA-ME analysis increased the size of enriched terms covering more proteins from the same processes than in MBR. It even enriched new terms that were present in our previous bulk analysis (cf. Supplementary Fig. 7H), leading to an overall set that was much more similar to that of the higher input sample. This finding complements our previous result that quantitative accuracy of additional identifications is maintained in DIA-ME analysis (Fig. 3C), therefore showing that their gain contributes the information retrieval to make biological inferences. Collectively, these data demonstrate that DIA-ME not only improves proteome coverage in samples of exceedingly low input, but especially that it allows quantification of proteins that underlie the biological process at stake. Of note, these proteins were quantified from a 1000-times lower input (200 pg) than the bulk experiment (200 ng), yet from a proteome whose coverage was only reduced by less than half (cf. Fig. 4B and Supplementary Fig. 7A).

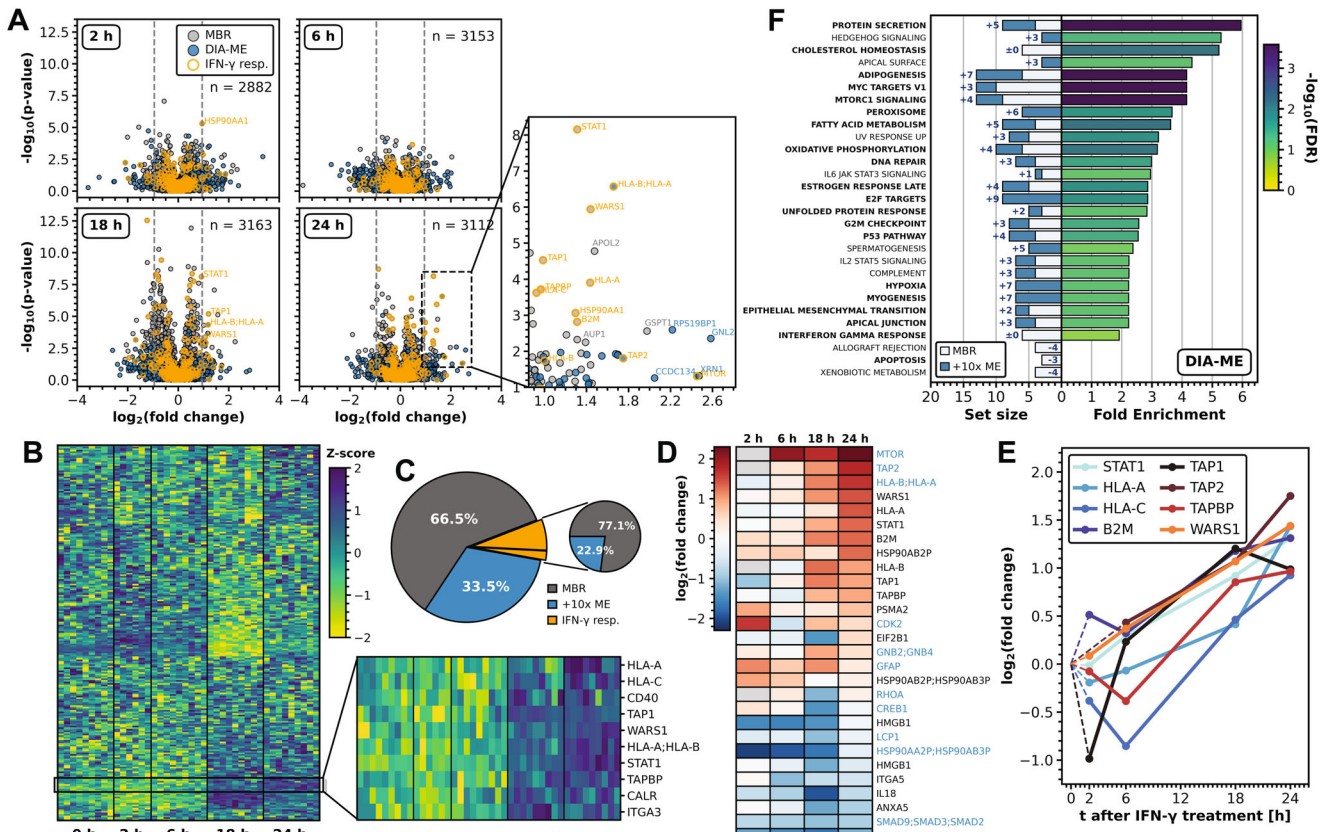

**Fig. 5 | Exploration of IFN-γ-induced immune response from the enhanced analysis of 200-pg samples. A** Volcano analysis of two-sided Student's t-test results from 200-pg time-point samples after IFN-γ treatment and the respective 0 h control. Number n of differentially expressed proteins indicated. Proteins whose expression could only be observed in DIA-ME analysis are highlighted in blue, proteins described in IFN-γ-response are encircled in yellow. **B** Heatmap of known IFN-γ-responsive proteins after hierarchical clustering by Euclidean distance. Missing values were imputed by k-nearest neighbors. Colors indicate the quantitative changes compared to the protein's median across all samples by Z-score. The black box outlines a cluster that shows gradual increasing Z-score over time. **C** Pie charts showing the proportional origin of differentially expressed proteins from **A**. Proteins already identified by MBR shown as dark grey wedges and exclusive identifications by DIA-ME shown as blue wedges. The small pie displays

the respective proportions for known IFN-γ-responsive proteins. **D** Heatmap of significantly up- and down-regulated proteins from panel A (p-value < 0.05, log₂ fold change > 0.95), showing their regulation over the course of treatment. Proteins indicated in blue were only found in DIA-ME analysis. **E** Line plot of selected proteins from **D**, showing their collective up-regulation over the course of treatment. **F** Gene set enrichment analysis of proteins up-regulated after 24 hours (log₂ fold change > 0.58) using MSigDB hallmarks. Bars on the right represent the enrichment degree, while their colors specify the enrichment's FDR. Bars on the left overlay the size of the enriched term and are depicted for MBR (white) and DIA-ME (blue) analysis with numbers indicating the additional contribution of DIA-ME. Bold terms indicated equivalent enrichment from the 200-ng bulk analysis (see Supplementary Fig. 7H). Source data are provided as a Source Data file.

## Single-cell proteomic analysis by DIA-ME reveals protein co-expression and co-existence of cell states

Observing the benefit of DIA-ME for low-input proteomics, we next aimed to assess its performance in the analysis of actual single cells. We therefore used 24 hours IFN-γ-treated and control U-2 OS cells, and obtained a total number of 143 individual cells by FACS sorting. In addition, we collected instances of 10 cells to serve as ME samples during subsequent data analysis (Supplementary Data 8).

According to our explanation of the operating principle of DIA-ME (Fig. 1A), a small number of MEs should provide sufficient peptide information to the search to analyze in theory an unlimited number of single cells. To verify this hypothesis, we analyzed different numbers of single cells with a constant number of 10-cell samples, and found consistent protein identifications per cell irrespective of how many cells were involved in the search (Supplementary Fig. 9A), confirming the scalability of the DIA-ME approach. When we involved all 143 cells, analogous to our findings from bulk samples, co-analysis with 10-cell samples improved the proteomic depth from 496 to 575 median protein groups per individual cell (+16%) (Fig. 6A) and identified a total of 1553 proteins (+41%) (Supplementary Fig. 9B). Although proteome coverage was modest on an absolute scale, as sample preparation was

conducted in-plate without further optimization, it fully served to demonstrate the benefit of DIA-ME, causing an expansion of the detected dynamic range in comparison to what would have been possible by conventional MBR analysis (Fig. 6B). In particular, DIA-ME also led to a higher peptide coverage per protein (Fig. 6C, D), facilitating their quantification and underpinning the advantage of this analysis for individual cells. Most important for the analysis of such low input amounts, we did not identify a single protein from the equivalent preparation of empty FACS droplets, nor in the blank runs in between single-cell measurements (Supplementary Fig. 9C), testifying the absence of contaminations and carryover in this experiment.

To investigate the proteome effect of IFN-γ and reduce the presence of missing values, we only retained data where at least 500 proteins per cell were identified after DIA-ME analysis (90 cells remained). Performing a differential expression analysis, we found that several of the IFN-γ-induced proteins observed in bulk and low-input samples were also significantly upregulated in single-cell data (Supplementary Fig. 9D). This comprised TAP1, TAPBP, HLA-A and HLA-C, which drove the cluster separation between untreated and treated cells in PCA analysis along the principal component 2 (Fig. 6E). Accordingly, these proteins displayed higher expression levels in

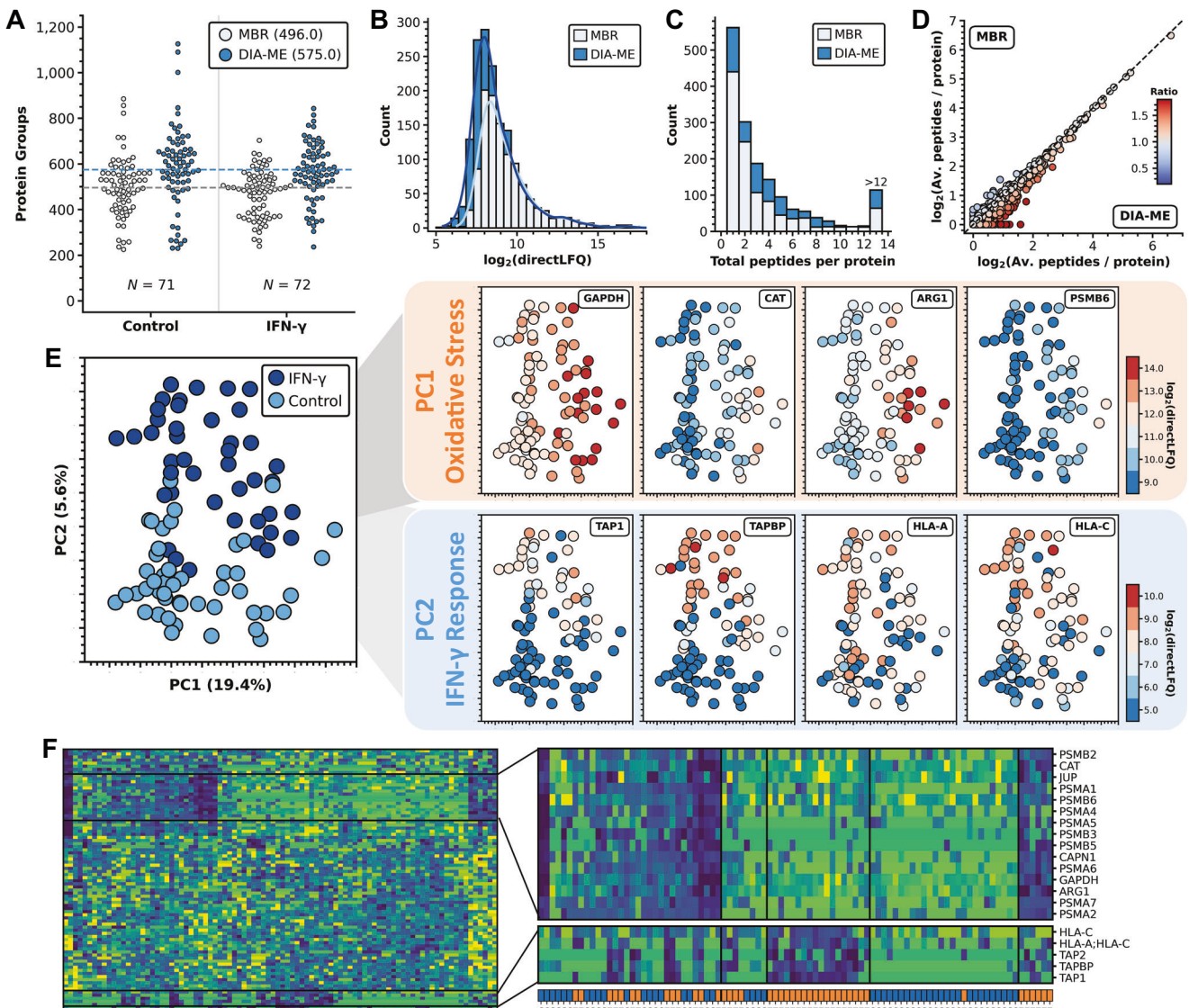

**Fig. 6 | DIA-ME-assisted analysis of individual U-2 OS cells reveals co-existence of metabolic states. A** Protein groups identified per individual cell in control (left) and 24 h IFN-γ-treated cells (right). Data were analyzed using conventional MBR (white) or DIA-ME (blue) using 10-cell MEs. The number *N* of single cells per condition are indicated. **B** Histogram of protein intensities for MBR (white) and DIA-ME analysis (blue). Curves calculated by Kernel density estimation. **C** Histogram of total identified peptides per protein for both analyses (white: MBR; blue: DIA-ME). **D** Scatter plot of log₂-transformed average peptide numbers per protein identified in MBR (y-axis) and DIA-ME analysis (x-axis). Colors represent the ratio of peptide identifications between the two analyses, effectively showing higher protein sequence coverage in MBR (blue) or in DIA-ME (red). Equal peptide numbers (ratio of 1) represented by a black line. **E** Principal component analysis (PCA) of control

and IFN-γ-treated cells after DIA-ME analysis based on known IFN-γ-responsive proteins (left). The two main principal components can be explained by the indicated processes on the right (Orange: PC1 – oxidative stress response proteins; Blue: PC2 – IFN-γ response proteins). Colors specify the degree of expression per cell ranging from blue (low) to red (high) for each of the indicated proteins. **F** Left: hierarchical cluster by Euclidean distance among individual cells (columns) and known IFN-γ-responsive proteins (rows). Colors in the heatmap indicate quantitative changes compared to the protein's median value across all cells by Z-score. Frames show two identified clusters (C1, C2), enlarged on the right. Color bar on the bottom indicates whether the column originates from a control (blue) or IFN-γ-treated cell (orange). Source data are provided as a Source Data file.

treated cells, however, we observed a notable disparity in protein expressions within each condition, resulting in an almost 10-fold range of abundances among treated cells in case of TAP1 and TAPBP (Supplementary Fig. 9E). Interestingly, protein levels in some of the cells were as low as in untreated cells, suggesting differences in their responsiveness, or that the attained expression levels depend on the initial level in the untreated cell.

To discern the distinct patterns of protein expression among the individual cells, we conducted a hierarchical clustering analysis. We found two protein clusters, one of which comprised structural and catalytical subunits of the 20S proteasome and metabolic enzymes like catalase (CAT) and GAPDH (cluster C1), while the other contained

proteins of the antigen presentation pathway, such as TAP1/2, TAPBP and HLA-C (cluster C2) (Fig. 6F). As expected from the previous analyses, cells in cluster C2 grouped together according to their experimental condition, showing upregulation of the respective proteins in treated cells, however, we observed that proteins in cluster C1 divided into two groups regardless of the treatment. Interestingly, the regulation of these proteins turned out to drive the principal component 1 of our initial PCA analysis (Fig. 6E) that constitutes the main process to separate the cells. Since these proteins are e.g. involved in oxidative stress response (GAPDH, CAT, ARG1) this observation indicates that cells with different metabolic activity co-exist in the initial cell population, yet such diversity does not dictate the responsiveness to IFN-γ.

Moreover, these data show that GAPDH differed 8-fold in expression between individual cells (Fig. 6E), calling into question its conventional status as a housekeeping protein. Indeed, it was found that GAPDH drastically differs in expression between cell types, both on protein[51] and transcript level[52], and has even been shown to be heterogeneously expressed among individual cells using single-cell RNA analysis[53,54].

As we aimed to further explore the existence and degree of protein co-expression within the same cell, we performed a co-variation analysis across proteins that were identified in ≥20 cells, irrespective of the treatment with IFN-γ. This has the potential to provide insight into the correlation in function or regulation of expression that cannot be obtained from bulk populations and that is unique to single-cell analysis. Among 570 × 570 protein pairs, this revealed two clusters that showed high correlation within each cluster while showing inversed correlation between them (Fig. 7A), encompassing 111 proteins (Supplementary Fig. 10A and Data 9). The strongest co-varying proteins (Supplementary Fig. 10B) include several complex-forming proteins, comprising the 20S proteasome subunit (PSMA4/5/6), the prohibitin complex (PHB1/2), ATP synthase F1 subunit (ATP5F1A/B), and the Hsp60-Hsp10 chaperonin complex (HSPD1/E1), but also functionally related proteins, e.g. the peroxiredoxins PRDX1/2, and the adherens junction proteins JUP and DSP (Fig. 7B). Moreover, we observed strong anti-correlation between various protein pairs of diverse function

(Supplementary Fig. 11), suggesting mutually exclusive expression. Overrepresentation analysis of the two clusters revealed complementary cellular functions, showing the specific enrichment of degradative processes in cluster C1, such as the proteasome, DNA damage response and apoptosis, and proliferative processes in cluster C2, including glucose metabolism, TCA cycle, regulation of the mitotic cell cycle and formation of ATP (Supplementary Fig. 10C). Given their inverse correlation, this suggests the co-existence of cells in two mutually exclusive metabolic states. Using network analysis, we further demonstrate that a hub of proteasome proteins drives the cluster harboring degradative processes, while the inversely related proteins that connect both clusters may provide interesting examples to infer novel functional relationships (Fig. 7C). For instance, GAPDH (glyceraldehyde 3-phosphate dehydrogenase) appeared to be the hub protein in the network linking the mutually exclusive association between specific proteins involved in glycolysis, TCA cycle, oxidative stress and mitochondrial activity. GAPDH is a crucial enzyme in glycolysis, however, has recently been described to divert glycolytic flux into the pentose phosphate pathway upon oxidation by intracellular hydrogen peroxide ($H_2O_2$)[55,56], making the cells tolerant to oxidative stress[56]. Our data may thus be explained by the interplay between these physiological processes including co-expression of GAPDH and PKM (pyruvate kinase M1/2) with oxidative response proteins, such as CAT

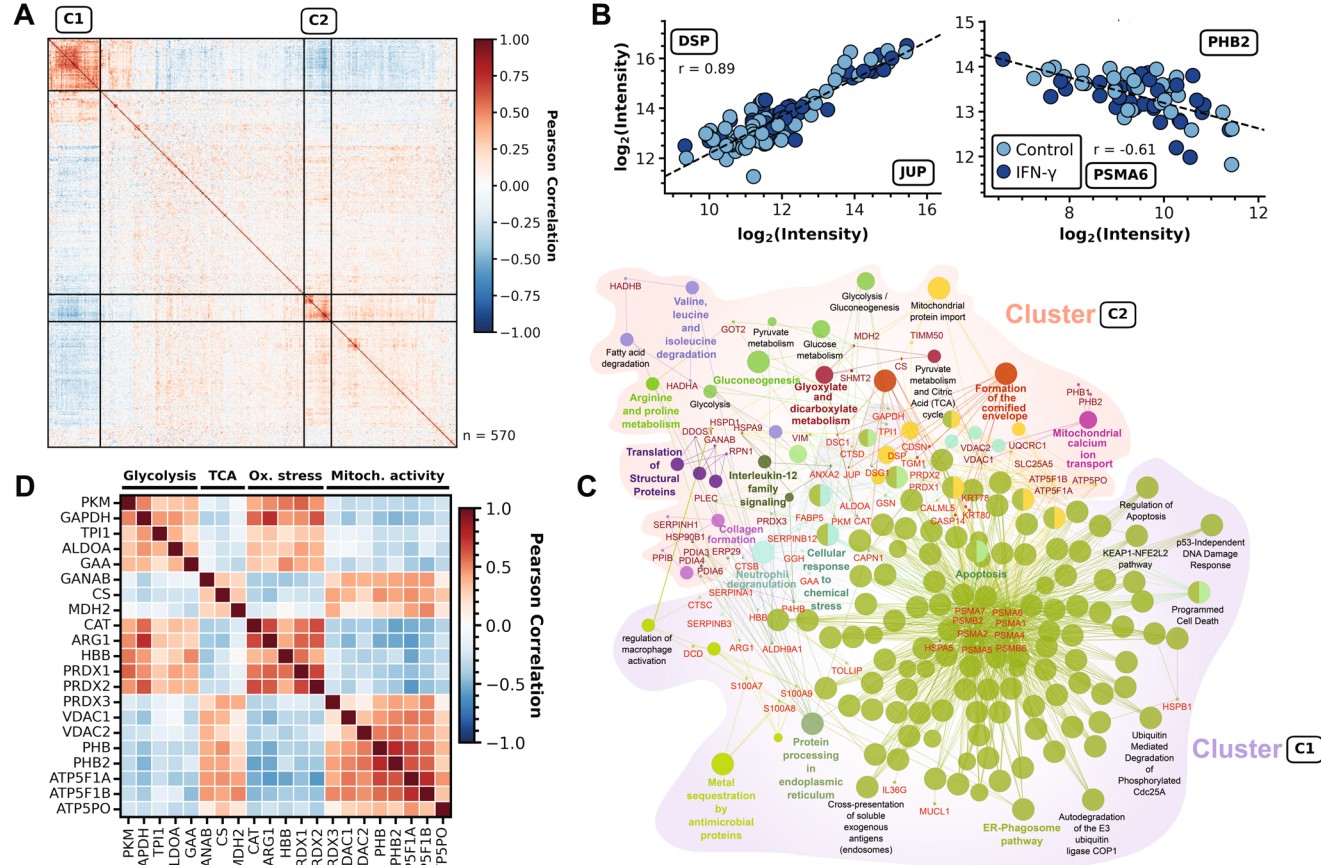

**Fig. 7 | Focused analysis of protein correlation modules in single cells. A** Co-expression analysis by Pearson correlation of proteins identified in ≥20 cells ($n = 570$). Two identified clusters (C1 and C2) with strong internal correlation but mutual anti-correlation are highlighted. **B** Individual expression levels per cell for pairwise positively correlated proteins JUP (x-axis) and DSP (y-axis) (left panel), and negatively correlated proteins PSMA6 (x-axis) and PHB2 (y-axis) (right panel). Colors indicate control (blue) and IFN-γ-treated cells (dark blue). Linear regression by Pearson shown as dashed line with the respective correlation factor r indicated. **C** Protein-protein interaction and protein-pathway interaction network, showing

relations within and between clusters C1 and C2 of **A**. The nodes of the network represent the terms associated with indicated proteins (Red: cluster C1; Dark red: cluster C2), and (undirected) edges represent interactions between proteins. Nodes represent significantly enriched pathways, while nodes of similar function are grouped by their color. **D** Co-expression analysis of proteins selected from (**C**) showing inversely correlated expression of metabolic proteins between cells. Proteins were assigned to processes by manual annotation as indicated. Source data are provided as a Source Data file.

and the peroxiredoxins PRDX1/2, resulting in reduced TCA and mitochondrial activity (Fig. 7D), and could be the consequence of detachment of cells, which is known to elevate endogenous oxidant levels[56–58]. In a related context, we observed that the expression of cytosolic peroxiredoxin (PRDX)-1 and 2 anti-correlated with mitochondrial PRDX3 (Fig. 7D), showing different modes of expression regulation of these isoforms, potentially related to their complementary functionality in their respective sub-cellular locations. Other examples are serine proteases SERPINA1, -B3 and -B12 that negatively correlate with SERPINH1 (Fig. 7C), suggesting mutually exclusive expression to possibly regulate collagen formation. In addition, we found that proteasomal proteins negatively correlate with several dozens of other proteins (Fig. 7C), potentially indicating enzyme-substrate relationships. These and many other observations from co-expression analysis exemplify hypotheses enabled by single-cell proteome analysis, opening exciting novel avenues to investigate causality of these interactions.

In conclusion, we applied DIA-ME to successfully explore proteome dynamics in single cells upon IFN-γ treatment, using 10-cell samples as matching resource to improve proteome coverage. The concept is readily scalable to larger cell numbers, and is extendable to any other cell types and treatments to investigate proteome response and protein co-variation in other biological contexts.

## Discussion

In this study, we introduced DIA-ME as an experimental approach to increase proteome coverage and completeness in low-input and single-cell proteomics. DIA-ME employs a higher-input reference sample that is co-analyzed with low input samples of interest, and was designed to exploit the capabilities of DIA analysis tools such as DIA-NN and Spectronaut. In particular, we refer to the reference sample as matching enhancer because it serves to generate an augmented internal library in DIA analysis software for improved matching to features from low-input data, resulting in an increased number of protein identifications. This approach is easy to implement in existing proteomic pipelines as it just requires the analysis of a small set of higher input amounts along with the samples of interest. We determined that an ME sample containing 10x higher input (e.g. 10 cells along with single-cell samples) suffices to considerably increase proteome coverage and data completeness. In this regard, DIA-ME is advantageous over library-based DIA applications that require time-intensive generation of spectral libraries. Furthermore, DIA-ME makes economic use of potentially scarce biological material since only a few ME samples need to be analyzed along with any desired number of low input samples. Conceptually even one ME sample would be sufficient to introduce an adequate number of additional features to the search, however, slightly more replicates could be used to account for potential missing values, as we did in this study. In this sense, MEs of extreme time-points during IFN-γ treatment resulted in higher proteome coverage than MEs of individual time-points, with equivalent results when all time-points were used (Supplementary Fig. 8A). This emphasizes that only a few MEs are required to cover the detectable peptide space, in theory along with a limitless number of single cell samples, making DIA-ME a highly scalable approach (Supplementary Fig. 9A). This contrasts with the concept of carrier channels in TMT-experiments, where a sample at 20–100× the amount of a single-cell proteome should be added to every plex of 8 or 16 single cells. Moreover, the amount needs be carefully tuned to account for ratio compression and collection of sufficient number of ions to allow for appropriate peptide quantification[19]. Thus, DIA-ME will be particularly advantageous in scenarios where cells of interest are scarce or hard to obtain, e.g. in clinical proteomics or developmental biology.

A crucial element of DIA-ME is its reliance on the feature matching functionality as implemented in DIA data analysis tools, and therefore we extensively evaluated the degree of false transfers, particularly in scenarios where low and high input samples are co-analyzed (Fig. 2). To this end, we used a two-proteome model and analyzed non-spiked and spiked samples together to assess the fidelity of the MBR function, as has been done previously for DDA in MaxQuant[59]. This revealed that the quality both of precursor identification and identity transfer was not affected in DIA-NN (FPR < 0.2% and FTR < 0.2%, respectively), even when co-analyzing a 1-ng human sample with a 100-fold excess ME sample that contains 20% (i.e. 20 ng) E.coli peptides. Counter-intuitively, FTRs even improved with increased size of the ME sample (Fig. 2C), which we hypothesize is the result of better spectral quality that enhances the matching process. Although we did not investigate this further, spectral data obtained from MEs may be much more similar to the data in low-input samples than to higher-input or even deep-fractionated libraries[60,61], which has been reported to promote false discoveries due to the large query space[62,63]. Indeed, we observed that the average number of proteins does not increase with extensive database size beyond 10x DIA-ME (Fig. 1B and Fig. 4B). Collectively, these data show that the matching process in DIA-NN is resilient to the presence of highly abundant interfering peptides, while effectively discriminating false- and true-positive identifications. DIA-NN out-performed Spectronaut in both aspects, which might be attributed to an affected FDR control (Fig. 2B) and the lenient default filtering in Spectronaut in case of low sample inputs and in the context of DIA-ME (Supplementary Fig. 5). Meanwhile, it will be interesting to identify the specific determinants in DIA-NN contributing to this observation, and to potentially improve its performance even further by tuning parameters especially for single-cell applications, e.g. with regard to the role of the TIMS dimension in diaPASEF data for the matching algorithm[27,64]. Until then, and as a practical implication, we found that the (non-default) run-specific q-value in DIA-NN offered high specificity discriminating false from true identifications (Supplementary Fig. 4B) and we therefore suggest filtering low-input data additionally at a moderate cutoff of e.g. 5%. Moreover, we recommend the use of 10x ME samples as they provide a good balance between increased proteome coverage and consistent identifications, even though the exact ratio may require further investigation.

With the notion that single-cell proteomics is still in its infancy, applications are diversifying from analytical studies that verify the ability to distinguish different cell types[18,30,32,33,65–68], to experiments that explore proteome changes induced by potent treatments such as LPS stimulation[35,69] or that investigate more subtle effects e.g. during cell cycle[34] or in stem cell populations[31]. Herein, we applied DIA-ME to investigate the proteome response to IFN-γ, which only induces a mild effect in U-2 OS cells as determined in a bulk experiment not limited by sample input (Supplementary Fig. 7). When using low-input sample amounts either from diluted cell extracts (200 pg peptides) or from single cells, DIA-ME recapitulated the main observations of the bulk experiment by quantifying the up-regulation of known IFN-γ response proteins, showing very similar time-resolved expression profiles (cf. Supplementary Fig. 7G and 5E) and enriched gene sets (cf. Supplementary Fig. 7H and 5F). This indicates that DIA-ME can detect biologically relevant consequences of mild treatments in single cells. Of note, these results were reliant on proper normalization to ensure unaffected protein quantification, where the recently published directLFQ algorithm[40] showed the best quantitative accuracy among several tested normalization strategies (Supplementary Fig. 6A). The ability of DIA-ME to study biological processes at the single cell level is likely to be more pronounced with increased proteomic depth, e.g. when using sample preparation protocols optimized for single cells, and using more sensitive mass spectrometry technologies than those used here.

Finally, in our data we observed many examples of proteins whose expression was co- or anti-correlated within the same cell, indicating the co-existence of different cell states within the population of U2-OS cells (Fig. 7 and Supplementary Fig. 10). The ability to obtain such

patterns of co-varying proteins is a major perspective of single-cell proteomics to elucidate fundamental aspects that underlies heterogeneity within cell populations, to reveal regulatory processes in protein expression, or to infer other functional relationships between proteins[70]. This concept has been successfully used in bulk proteomics to reveal novel functions of uncharacterized proteins[71], but it has greater potential when performed at the single-cell level, which does not suffer from masking effects due to heterogeneity of cell populations. Indeed, this has begun to be explored in recent single-cell studies, observing co-regulation of proteins that can be rationalized from their interaction in complexes[72] or their complementary involvement in energy metabolism[66]. In our data, we observed many instances of proteins with inverse expression patterns, but with similar functions (e.g. in glycolysis, oxidative stress, proteases), suggesting mutually exclusive or complementary functionality. Although our data cannot infer causality of these relationships, they provide novel hypotheses that can be tested in future studies. From a systems perspective, it will be interesting to understand if modules of co- and anti-correlated proteins are maintained across different states of the same cell type, or across different cell types. DIA-ME should provide a powerful approach to conduct such studies in the future to increase our understanding of proteome regulation at the single-cell level.

## Methods

### Preparation of *E.coli* peptides
Lyophilized *E.coli* K12 sample (Bio-Rad) was reconstituted in 50 mM triethylammonium bicarbonate (TEAB, pH 8.5, Sigma-Aldrich) buffer to a stock concentration of $1 \mu g/\mu L$. $100 \mu g$ were incubated at $95\,°C$ for 5 min and subsequently worked-up by SP3[73]: a mixture of 50:50 carboxylate-modified Sera-Mag SpeedBeads type A and B (Cytiva) were washed three times with ddH$_2$O (Barstead GenPure, Thermo Scientific) on a magnetic rack and 1 mg of combined beads were added to the *E.coli* sample. For protein aggregation, the suspension was filled up with acetonitrile (ACN, Biosolve Chimie) to a concentration of 75% (v/v) and incubated at RT and 800 rpm for 20 min. Afterwards beads were rinsed twice with 1 mL 80% (v/v) ethanol (EtOH, VWR) and once with $800 \mu L$ ACN. During washing steps the suspension was homogenized to ensure adequate bead-solvent interaction. In the end, beads were air-dried for 2 min to get rid of ACN leftovers and resuspended in $30 \mu L$ digestion buffer (50 mM TEAB, pH 8.5 + 2 mM CaCl$_2$ (Sigma-Aldrich)). Disulfide-bonds were reduced by 10 mM final concentration dithiothreitol (DTT, Biomol) and incubation at $37\,°C$ and 600 rpm for 45 min. Next, bare cysteine residues were alkylated with 55 mM final chloroacetamide (CAA, Sigma-Aldrich) at RT and 600 rpm in the dark for 30 min. Finally, proteins were digested overnight at $37\,°C$ and 800 rpm using sequencing-grade modified trypsin (Promega) at a protein-enzyme-ratio of 50:1. On the next day, the supernatant was transferred to a fresh tube and the beads were additionally washed with $50 \mu L$ 1% (v/v) trifluoroacetic acid (TFA, Biosolve Chimie) in ddH$_2$O at RT and 800 rpm for 5 min to improve peptide recovery. The obtained peptide solution of pooled supernatant and washing fraction was subsequently cleaned-up on SepPak cartridges (Waters). The column was prepared according to the instructions of the manufacturer using 1 mL of ACN, solvent B (50% (v/v) ACN in ddH$_2$O + 0.1% (v/v) formic acid (FA, Biosolve Chimie)) and solvent A (ddH$_2$O + 0.1% (v/v) FA), respectively. Peptides were loaded on top of the column and washed twice with 1 mL solvent A before being eluted twice in $200 \mu L$ solvent B. Purified peptides were frozen at $-80\,°C$, lyophilized in a freeze-dryer and afterwards reconstituted in solvent A. The resulting peptide concentration was determined in a colorimetric assay using bicichoninic acid (BCA) (Pierce, Thermo Scientific).

### Generation of two-species proteome model
Commercial HeLa S3 protein digest (Pierce, Thermo Scientific) was reconstituted in ddH$_2$O + 0.1% (v/v) FA by gentle vortexing. For

preparation of spiked human samples, defined amounts of the HeLa peptide standard were mixed with 5%, 10% and 20% (w/w) of prepared *E.coli* peptides to final concentrations of $0.5 \text{ ng}/\mu L$ ($N = 7$), $2.5 \text{ ng}/\mu L$ ($N = 3$), $5 \text{ ng}/\mu L$ ($N = 3$) and $50 \text{ ng}/\mu L$ ($N = 3$) per spike ratio. In addition, pure HeLa peptide standard was diluted in ddH$_2$O + 0.1% (v/v) FA to a final concentration of $0.5 \text{ ng}/\mu L$ ($N = 7$) (human-only).

### Cultivation of U-2 OS cells and Interferon-gamma treatment
Human osteosarcoma epithelial cells (U-2 OS cell line) were cultivated in DMEM high glucose medium (Gibco), supplemented with 2 mM L-glutamine, 10% (v/v) fetal bovine serum (Gibco), and an additional 2 mM GlutaMAX (Gibco). For the exploration of interferon gamma (IFN-γ) stimulation effects, U-2 OS cells underwent treatment with 50 ng/mL recombinant IFN-γ (Cell Signaling). The cytokine was diluted in 0.5% (w/v) bovine serum albumin (BSA, Serva).

In the context of preparing the time series experiment, cells were cultured in 10-cm dishes in three biological replicates, and IFN-γ treatments were administered when the cell confluence reached 70%. Cells were kept with the treatment for durations of 2 hours, 6 hours, 18 hours, and 24 hours. Additionally, distinct 0-hour and 24-hour time points were collected as controls.

After the respective incubation periods, the culture medium was carefully aspirated, and the cells underwent two rounds of washing with pre-warmed phosphate buffered saline (PBS, Sigma-Aldrich). Adherent cells were detached by gentle scraping in 5 ml of PBS, and subsequently centrifuged at 500 x g for 10 min. The resulting cell pellets were collected, rapidly frozen, and stored at $-80\,°C$, preserving their integrity until their following preparation.

### Preparation of proteomic samples
For bulk analyses of 200-pg and 200-ng injections, U-2 OS cells obtained from cell culture ($N = 3$) were thawed and lysed in 50 mM TEAB, pH 8.5 (Sigma-Aldrich) buffer + 2% (v/v) Sodium dodecyl sulfate (SDS, Bio-Rad). Proteins were denatured at $95\,°C$ and 600 rpm for 10 min before adding a final concentration of 2% (v/v) TFA (Biosolve Chimie) to the lysate. The reaction was quenched after 1 min by neutralizing pH with 3 M tris(hydroxymethyl)aminomethan (Tris, AppliChem) solution. Afterwards the lysate was sonicated for 15 cycles of 30 s ON/OFF at $10\,°C$ using a Bioruptor (Diagenode) and subsequently frozen at $-80\,°C$. Prior to protein clean-up, the sample was thawed, centrifuged at 18,000 x g to separate remaining cell debris and transferred to a fresh tube. A total of $20 \mu g$ protein per sample (determined by BCA, Pierce Thermo Scientific) were employed in the SP3 protocol described above using $200 \mu g$ Sera-Mag SpeedBeads (Cytiva) and $200 \mu L$ of respective washing solutions. After digestion and subsequent bead washing, the acidified peptide solution was cleaned-up on self-packed StageTips[74] accommodating four discs of Empore C18 material (Merck Supelco). The material was prepared by consecutive application of pure ACN (Biosolve Chimie), solvent B (50% (v/v) ACN + 0.1% (v/v) FA (Biosolve Chimie) in ddH$_2$O) and solvent A (0.1% (v/v) FA in ddH$_2$O). Peptides were loaded on top of the column and washed twice with 1 mL solvent A, followed by elution in $100 \mu L$ solvent B. Purified peptides were frozen at $-80\,°C$, lyophilized in a freeze-dryer and afterwards reconstituted in solvent A. The resulting peptide concentration was determined by BCA assay (Pierce, Thermo Scientific).

### FACS sorting and preparation of individual cells
For our single-cell proteomics experiment, we utilized 24 hours control and IFN-γ-treated U-2 OS cells from the previously obtained pellets. Cells were prepared by gentle trypsin digestion (0.25%) to establish a homogeneous population of singularized cells. Approximately one million cells were diluted in 1.5 mL PBS (Sigma-Aldrich) and immediately administered to the cell sorting via a BD FACSAria III instrument (BD Biosciences), utilizing a 384-well plate configuration. Default settings for optical filters and mirrors were employed to

facilitate the detection of the scattered signals. Parameters such as laser power, PMT-voltage settings, and gating were maintained at constant level throughput the entire experiment. Single cells were subjected to the sorting event using forward and side gating, while non-viable and dimerized cells were excluded (FACS gating shown in Supplementary Fig. 12). The sorting procedure was conducted at ambient room temperature. Sorted single and ten cells were directly collected in 2 µL of lysis buffer (50 mM TEAB, pH 8.5 + 0.025% (w/v) n-dodecyl-β-D-maltoside (DDM) (Sigma-Aldrich)) per well, immediately transferred to a chilled environment (ice-box), and subsequently prepared by the following in-plate protocol: cellular proteins were denatured at 70 °C and 600 rpm for 30 min, before adding 2 µL of 0.5 ng/µL trypsin (Promega) in digestion buffer (50 mM TEAB, pH 8.5) to the wells. Samples were incubated at 37 °C and 600 rpm for 1 h and afterwards acidified by 2 µL 0.5% (v/v) FA. The resulting volume of around 5 µL was transferred to a 96-well autosampler plate and subsequently injected into MS.

## Data acquisition (LC-MS/MS)

Injections were performed in seven technical replicates for 1-ng samples (spiked and non-spiked) and three technical replicates for 5-ng, 10-ng and 100-ng spiked ME samples in the DIA-ME evaluation experiment (total of 55 injections). For the 200-pg IFN-γ experiment, time-point samples[6] of individual cell cultures ($N = 3$) were each injected in three technical replicates, while 1-ng, 5-ng and 50-ng ME samples were injected once per time-point (total of 108 injections). The bulk 200-ng IFN-γ experiment comprised four time-points from three individual cell cultures, respectively, which were injected in one technical replicate (total of 12 injections).

Desired peptide amounts were injected in 2 µL volume onto an analytical column (IonOpticks Aurora Series, 25 cm × 75 µm i.d. + CSI, 1.6 µm C18) using an EASY-nLC 1200 system (Thermo Scientific). Peptides were separated across a 15 min active gradient starting from 3.2% (v/v) ACN concentration in ddH$_2$O (Biosolve Chimie) + 0.1% (v/v) FA to 13.6% (v/v) in 7.5 min to 20.0% (v/v) in 3.5 min to 28.0% (v/v) in 4 min at a flow rate of 300 nL/min and a temperature of 50 °C maintained by a column oven (Sonation). The LC system was connected to a timsTOF Pro mass spectrometer (Bruker Daltonics) via a nano-flow electrospray ionization (nano-ESI) source (Captive Spray, Bruker Daltonics). Analytes were ionized at 1,500 V capillary voltage, 3.0 L/min dry gas and 180 °C dry temperature. MS data was acquired in diaPASEF mode. In the TIMS, ions were accumulated to an IM constant $1/K_0$ of 1.7 V*s/cm$^2$ and sequentially ramped from 1.3 to 0.75 V*s/cm$^2$ over 100 ms in a locked duty cycle. Subsequent MS1 scans were performed from 200 to 1,700 m/z, while only precursors with a mass ratio of 475 to 1,000 m/z were considered for DIA window isolation. The range of 525 m/z was covered by equally sized windows of 25 Th width and 0.15 V*s/cm$^2$ heights, which were combined to a total of 8 DIA scans (Supplementary Fig. 1A) and resulted in 0.95 s cycle time. Precursor fragmentation was induced by IM-dependent collisional energies from 45 eV at $1/K_0$ of 1.3 V*s/cm$^2$ to 27 eV at 0.75 V*s/cm$^2$. The following ion detection was performed in high sensitivity mode for samples containing ≤50 ng peptide amount. In case of the analysis of single cells ($N = 143$) and 10-cell ME samples ($N = 15$), data was acquired from an injection volume of 5 µL (entire sample) using the described method, but utilizing a flow rate of 200 nL/min (total of 158 injections).

The analysis of conventional bulk samples (200 ng) was performed on an Orbitrap Fusion Tribrid MS (Thermo Scientific) connected to an EASY-nLC 1200 system (Thermo Scientific). This LC system was configured in a trapping setup, comprising a Acclaim PepMap 100 trapping column (2 cm × 100 µm i.d., 5 µm 100 Å C18, Thermo Fisher Scientific) and a consecutive nanoEase M/Z Peptide BEH analytical column (25 cm × 75 µm i.d., 1.7 µm 130 Å C18, Waters). Peptides were loaded onto the trapping column at constant pressure of 800 bar, utilizing a total volume of 22 µL ddH$_2$O + 0.1% (v/v) FA.

Subsequently, peptides were separated in the analytical column along a 87 min gradient starting from 2.4% (v/v) ACN concentration in ddH$_2$O + 0.1% (v/v) FA to 6.4% (v/v) in 4 min to 8.0% (v/v) in 2 min to 25.6% (v/v) in 68 min to 40.0% (v/v) in 12 min to 80% (v/v) in 1 min at a constant flow rate of 300 nL/min and a temperature of 45 °C maintained by a column oven (MonoSLEEVE, Analytical Sales and Services). Peptides were introduced into MS via a Nanospray flex ion source (Thermo Fisher Scientific) utilizing a Sharp Singularity nESI emitter (ID = 20 µm, OD = 365 µm, L = 7 cm, α = 7.5°, Fossiliontech) connected to a SIMPLE LINK UNO-32 (Fossiliontech). The emitter maintained a spray voltage of 2.5 kV, and the ion transfer tube capillary temperature was set to 275 °C. Data was acquired in DDA mode using MS1 full scans between 350 and 1500 m/z at a resolution of 120,000, and a maximum injection time (IT) of 32 ms with automatic gain control (AGC) of $3.0 \times 10^6$. The top 20 most abundant precursors were isolated for fragmentations utilizing an isolation window of 2.0 m/z in the quadrupole, only allowing determined charges of 2–5. Dynamic exclusion was to set 40 sec with a mass tolerance of ±10 ppm. For MS2 acquisition, higher-energy collisional dissociation (HCD) was employed at 25% and the resulting fragments were acquired with an AGC target of $1.0 \times 10^5$ or a maximum IT of 50 ms, covering a scan range from 200 to 2000 m/z in centroid mode.

## Raw data processing

After MS acquisition, DIA raw data was processed using library-free analysis in DIA-NN 1.8.0 or Spectronaut 18. We used a UniProtKB sequence table (fasta-file) of *Homo sapiens* (downloaded in September 2021, 20,386 reviewed proteins) for in-silico digestion and added an *Escherichia coli* K12 database (downloaded in April 2022, 4,529 reviewed proteins) in case of *E.coli*-spiked data. Configurations and compositions of DIA raw data analysis are summarized in Supplementary Data 1.

For analyses in DIA-NN, we used deep learning-based prediction of spectra, retention times and ion mobilities and allowed a maximum of two missed cleavage sites (one for single-cell analysis) per peptide. We set peptide length range to 7–50 and precursor charge range to 2–4. Protein isoforms were grouped according to their protein names from fasta-files and we selected "any LC (high precision)" for precursor quantification. In case of the analysis of single cells, we moreover set the algorithm's scan window to a value of 5 and MS1/Mass accuracies to 1.5e-05. For DIA-ME experiments, we combined raw files from low-input samples and MEs in the search and activated the match-between-runs (MBR) function. Analyses that are indicated as "MBR" did not contain ME samples, however, also have the MBR setting activated, whereas in analyses indicated as "Indiv", raw files were searched individually without allowing feature matching. All other settings remained default, including data filtering at 1% precursor FDR.

For DIA-ME analyses in Spectronaut, we equivalently combined raw files from low-input samples and MEs in the search. Since Spectronaut matches features among samples by default, "MBR" refers to a search with unchanged settings without the addition of ME samples.

Analysis of DDA bulk data was performed in MaxQuant (v2.0.3.0) using a canonical *Homo sapiens* database (downloaded in November 2021, 20,394 reviewed proteins). Trypsin was specified as digestion enzyme, allowing up to 2 missed cleavage sites. Variable modifications included Oxidation (M), Acetyl (Protein N-term), and deamidation (NQ), while carbamidomethyl (C) was set as a fixed modification. For protein identification, the minimum unique peptides were set to 1, and peptide and protein hits were filtered at a 1% false discovery rate (FDR), with a minimum peptide length of 7 amino acids. The reversed sequences of the target database served as decoy for FDR calculation. The second peptide search option was activated. The MBR function was enabled with a matching time window of 0.4 min and an alignment time window set to 20 min. The "dependent peptides" function was deactivated. We activated label-free quantification and utilized unique

and razor peptides for quantification. All other MaxQuant settings were maintained at their default configurations.

## Data analysis

Analysis of proteomic data was performed using Python code (version 3.9.12) in the Jupyter Notebook environment (v6.4.8)[75] that can be accessed via a GitHub repository (https://github.com/krijgsveld-lab/DIA-ME). We relied on the Pandas (v1.4.2) and NumPy (v1.21.6) packages for data handling, and used the Matplotlib (v3.5.1) and Seaborn (v0.11.2) packages for data plotting. Further statistical calculations, such as Student's *t*-tests, hierarchical clustering and Gaussian approximation to model extracted ion chromatograms, were conducted by the SciPy (v1.7.3) package, while ROC curve calculation, dimensional reduction and data imputation were performed using Scikit-learn (v1.0.2). Specifically, we computed ROC data for all relevant q-values in Spectronaut and DIA-NN. Moreover, we performed the UpSet analysis using the UpSetPlot package (v0.6.1). For normalization of DIA-NN data, we employed the directLFQ package (v0.2.8)[40] after excluding ME samples from the report table and performing default filtering on precursor level using run-wise (Q.Value ≤ 0.01) and global q-values (Lib.Q.Value ≤ 0.01) and on protein level using the global q-value (Lib.PG.Q.Value ≤ 0.01). Furthermore, to evaluate the performance of data normalization, we applied the DiaNN (v1.0.1)[26] and iq (v1.9.6)[39] packages in R Studio using the same filter criteria and compared it to results from directLFQ. Gene set enrichment analysis (GSEA) was conducted utilizing the R package clusterProfile (v3.12.0)[76] based on the "fgsea" algorithm. Finally, illustrations in Figs. 1A, 1B, 4A and Supplementary Fig. 6C were created with BioRender.com, released under a Creative Commons Attribution-NonCommercial-NoDerivs 4.0 International license.

For the assessment of erroneous feature assignment in DIA-NN and Spectronaut, we used the following calculations per replicate of non-spiked HeLa proteome (peptides that are present in *H.sapiens* and *E.coli* proteins were excluded from the analysis):

False-positive rate:

$$\text{FPR} = \frac{\text{IDs}_{E.coli}}{\text{IDs}_{\text{Total}}} \quad (1)$$

with *E.coli* identifications $\text{IDs}_{E.coli}$ and total identifications $\text{IDs}_{\text{Total}}$.

False-transfer rate:

$$\text{FTR} = \frac{\text{IDs}_{E.coli,\text{MBR}} - \text{IDs}_{E.coli,\text{Indiv}}}{\text{IDs}_{\text{Total,MBR}} - \text{IDs}_{\text{Total,Indiv}}} \quad (2)$$

with *E.coli* identifications with and without activation of "MBR" $\text{IDs}_{E.coli,\text{MBR}}$ and $\text{IDs}_{E.coli,\text{Indiv}}$, and total identifications with and without activation of "MBR" $\text{IDs}_{\text{Total,MBR}}$ and $\text{IDs}_{\text{Total,Indiv}}$.

For the similarity evaluation of DIA-ME and MBR data sets, we used following calculations:

Overlap (Szymkiewicz-Simpson) coefficient

$$\text{overlap}(\text{MBR,DIAME}) = \frac{|\text{MBR} \cap \text{DIAME}|}{\min(|\text{MBR}|, |\text{DIAME}|)} \quad (3)$$

Jaccard similarity coefficient:

$$J(\text{MBR,DIAME}) = \frac{|\text{MBR} \cap \text{DIAME}|}{|\text{MBR} \cup \text{DIAME}|} \quad (4)$$

For the single-cell dataset, we performed co-variation analysis by calculating the Pearson correlation between all identified proteins, tolerating a minimum of 20 pairwise observations (Supplementary Data 9), and clustering the resulting correlations based on Euclidean distance. We identified two clusters and subsequently created a protein-protein interaction network using the Cytoscape software

(v3.9.1)[77], leveraging the KEGG database for pathway information and the ClueGO plug-in for functional annotation. The interaction network was constructed by importing the protein list of both co-expression clusters and enriching it with relevant pathway information utilizing the KEGG database. Further, we employed the ClueGO plug-in to functionally annotate and highlight enriched pathways within the network. To ensure robustness of the enrichment procedures, a significance threshold (p-value ≤ 0.01) was applied. The resulting network was directly visualized in Cytoscape.

## Data annotation, filtering and imputation

For the analysis of IFN-γ-treatment experiments (dilution and single-cells), we filtered the protein intensity table on a contaminants list that we extracted from MaxQuant (v2.0.1). Afterwards we annotated proteins when they were previously described to be responsive to IFN-γ. This list was retrieved from STRING database (https://string-db.org/) using IFN-γ-related signaling keywords, and extracting protein-protein interactions with confidence > 0.7 interaction score[78]. Likewise, the Gene Ontology Consortium (http://geneontology.org/) provided annotations for relevant molecular functions and biological processes related to IFN-γ signaling. The extracted data from both sources were integrated based on common identifiers, and proteins identified in both databases were cross-referenced to ensure consistency (Supplementary Data 7).

For the 200-pg dilution experiment, the obtained protein data was filtered on proteins that were identified in at least three replicates per time-point and imputed the remaining missing values using the k-nearest neighbors function before conducting hierarchical clustering. Furthermore, principal component analysis was performed solely on proteins that showed full data completeness.

Following the DIA-ME analysis of single-cells, we determined the number of protein groups per individual cell and filtered for those cells that showed more than 500 identified proteins. The remaining data was used for the differential expression and co-variation analysis. Moreover, we excluded proteins that were identified in less than 30% of cells for principal component analysis and hierarchical clustering of known IFN-γ-responsive proteins and imputed missing values by the lowest intensity value present in the dataset.

## Reporting summary

Further information on research design is available in the Nature Portfolio Reporting Summary linked to this article.

## Data availability

The acquired raw LC-MS/MS data and processed report files generated in this study have been deposited in the ProteomeXchange Consortium via the PRIDE partner repository under the accession codes PXD053462 (*E.coli*-spiked experiment), PXD048162 (bulk IFN-γ experiment), PXD053473 (low-input IFN-γ experiment) and PXD053464 (single-cell experiment). In addition, we uploaded SDRF metadata with our evaluation data set (*E.coli*-spiked experiment). All data generated in this study are provided in the Supplementary Information and in the Source Data file. Source data are provided with this paper.

## Code availability

All relevant code used for the analysis of data in this work is stored in a GitHub repository that can be accessed via https://github.com/krijgsveld-lab/DIA-ME.

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

## Acknowledgements

This work was funded in part by the Federal Ministry of Education and Research (BMBF) in the framework of MSCoreSys (project number: 161L0213), and by the Ministry of Science Baden-Württemberg within the framework of the Excellence Strategy of the Federal and State Governments of Germany. We specially thank the scOpenLab of the German Cancer Research Center (DKFZ) headed by Jan-Philipp Mallm for their help with FACS sorting of single cells. For the publication fee we acknowledge financial support by Heidelberg University.

## Author contributions

K.K. formulized the DIA-ME concept; K.K. and S.A.A. designed and performed the experiments; K.K. analyzed the data in discussion with S.A.A.; K.K. visualized the data; K.K. and J.K. drafted the manuscript; K.K., S.A.A. and J.K. revised the manuscript; J.K. supervised the work.

## Funding

## Competing interests

The authors declare no competing interests.
