## [Peer Review File · Nature Communications]

Reviewers' Comments:

Reviewer #1:

Remarks to the Author:

Krull et al. present benchmarks of a single-cell proteomics data analysis strategy, referred to as DIA-ME, and explore its utility in assessing the impact of IFN-gamma treatment on the proteome.

Major issue:

The primary concern with the study is that the use of MBR to increase proteome coverage in single-cell analysis is not a new concept (PMID: 35810282). Nonetheless, it is important to acknowledge that the benchmarks established by Krull et al. are among the most comprehensive to date. Given this, I would like to propose several enhancements:

1. It needs to be demonstrated that DIA-ME surpasses the performance of a GPF-based library.
2. The validation of DIA-ME by spiked sample co-analysis was performed using 1 ng injections. Can the authors demonstrate that their method is also reliable when applied to single cells?
3. The scalability of DIA-ME depending on the number of single cells analyzed needs to be investigated.
4. Do the extra IFN-gamma responders detected with DIA-ME using 200 pg injections show the same changes in the 200 ng experiment?

Minor comments:

- I could not find any information on how multiple testing correction was performed.
- Given the use of MBR, the traditional entrapment FDR calculation is no longer applicable. This explains the high values noted for Spectronaut. Error control should therefore be performed only using the FPR and FTR.
- I recommend that the authors include a supplementary table detailing the full configurations of the software used in their analysis.
- Why are intensity sums the same between 2 and 5 ng on Figure S1C?

Reviewer #2:

Remarks to the Author:

The manuscript by Krull and coauthors presented a detailed evaluation of the MBR feature of DIA-NN and Spectronaut software for low-input proteomics, with preliminary data for single-cell proteomics. While many evaluations have been done for bulk-scale DIA analysis, it has great value to make such a study focus on low-input

datasets, given the growing interest in single-cell analysis nowadays. The authors used two-species proteomes to study the quantitative precision, as well as false discovery rates at different input levels. The optimized workflow was used to study interferon-gamma treatment of tumor cells. While the data present here is a valuable resource for the field, the overall innovation is low. The reviewer doesn't believe it should be published in NC.

1. The use of high-input DIA samples analyzed together with low-input samples was widely used in the field and widely appeared in the vendor's technical report. The major contribution of this manuscript is the optimization of sample sizes for high-input samples. They also suggest normalizing the data using the directLFQ package to improve the precision. However, overall such improvement is incremental.

2. There is no new software or new algorithms were developed in this work. The DIA-ME essentially is a data analysis workflow, which doesn't justify being highly innovative.

3. Figure 5F, the correlation map indicates the correlation between 0 hours and 24 hours is higher than 18 hours. More surprisingly, at each time point, only 1-2 replicates showed low correlations with 0 hours, and there are sub-clusters in each time point, which are very confusing. The authors should investigate why the cell response is so diverse at each time point, giving these are bulk-scale (diluted) experiments.

4. The data quality of single-cell is very poor. There is no blank controls for this experiment (blanks mean empty FACS droplet with all sample prep procedures). The author should include such blanks to demonstrate the data is actually from single cells, as many recent data shows that background contamination is a major problem of single-cell proteomics

5. Figure 7A. The improvement of DIA-ME is relatively moderate for single-cell data. Indeed, PCA analysis indicates the two cell state only resolved at PC2 space, which indicates the quantitative precision is low for the overall workflow.

6. The author did two studies of the interferon-gamma treatment (Figure 5-7). Returning to the value of single-cell proteomics, did the authors identify new functional modules missed in bulk-scale (diluted) experiments? What's the biological significance of performing the single-cell assay if you can get the same or better answer from the bulk assay?

Minor comments:

1. Figure 4 B and 4C, the authors used different colors to indicate different samples in

one plot. However, everything is mixed. It is very challenging to identify the data from each sample. It is suggested to separate them into different plots

2. Figure 6F, there is some disagreement between MBR and DIA-ME. The author should discuss why some enrichments were only observed in MBR. There are several 0 and negative counts in MBR data, which should be clarified.

Reviewer #3:

Remarks to the Author:

The authors describe an improved mass spectrometry data acquisition method for single cell proteomics. In short, by providing proteomics search engines with additional data from a higher-sample input run, the identification efficiency can be improved through the search engine's 'match-between-runs' strategy. This method, coined DIA-ME (matching enhancer), is optimized and benchmarked on technical spike-in samples, and then evaluated on biological samples.

While the authors provide an extensive dataset to evaluate the effectiveness of the method, the current work lacks an in-depth analysis of the match-between-runs implementations in the used search engines. Indeed, the two closed-source tools that are used by the authors (DIA-NN and Spectronaut) behave very differently across the testcases, but the cause of these differences are not thoroughly investigated. Addressing the comments below could drastically improve the quality of the manuscript in terms of data analysis. I did not fully assess the acquisition and biological use case sections of the manuscript, as these are somewhat outside of the scope of my expertise.

1. The differences between DIA-NN and Spectronaut should be further investigated. The authors allude to a less stringent filtering in its 'first search' for Spectronaut. Additionally, MBR strategies are originally meant to be used between runs for similar samples. In this case, the ME run can be regarded as very different from the low-input sample runs. Therefore, depending on how FDR filtering is applied by the search engines for the initial search (on a per-run basis, or across all runs), ill-estimated FDR thresholds could lead to either too many false identifications or a decrease in identification sensitivity. Unfortunately, DIA-NN and Spectronaut are closed-source and can therefore be regarded as black boxes. Nevertheless, both (potential) issues could be easily investigated by first searching the ME runs separately, extracting the resulting spectral libraries, and then searching the low-input samples with these spectral libraries. Applying this additional test for both search engines, might shed some light on the

reported differences.

2. The authors consistently report results at the level of protein-group. While this is useful from a biological perspective, the search engines are identifying and transferring precursor identifications. For more concrete comparisons, and to avoid any biases from steps such as protein inference, I therefore strongly recommend reporting precursor level identification rates, false discovery rates, and false transfer rates. Especially for reporting the latter, precursor-level data would allow the authors to correctly account for the amount of *E. coli* vs human peptides in each database instead of the number of proteins, which does not account for differences such as protein lengths.

3. The authors benchmark the method on low-input samples instead of actual single cells. The reported numbers of identified protein groups are significantly lower for the actual single cell experiments (~500) than on these low-input samples (~2500). To address potential concerns on the transferability of the low-input sample benchmarks to actual single cell experiments, the authors could relatively easily assess the false transfer rates by searching the single cell runs with the spike-in ME runs.

4. From communications with colleagues, it seems that the acquisition of a bulk sample along single cell samples for use with DIA search engine MBR methods is already widely applied. It therefore seems that the authors have somewhat overstated the novelty of this approach in the manuscript. I would therefore recommend positioning the manuscript as an optimization and validation of the approach rather than an introduction of a novel method.

5. All scripts and source code for the work's data analysis seems to be missing from the manuscript submission. Sharing the Jupyter notebooks (for Python code) and R scripts could improve the clarity of the applied methods (e.g. calculation of the false transfer rate) and is a must for reproducibility of the work. Secondly, the raw proteomics data could prove to be a highly valuable resource to improve DIA-ME data analysis methods. Therefore, uploading SDRF metadata files alongside the raw data would strongly facilitate further development by the community. Thirdly, the authors seem to have only uploaded mzML data to PRIDE. For full reproducibility of the work, the unprocessed raw mass spectrometry files (.raw or .d) should be uploaded.

6. The manuscript contains many figures of which some only add minor value. I would suggest somewhat limiting the figures in the main manuscript to enhance readability, interpretability and minimize redundancies.

Rebuttal

Reviewer #1

Krull et al. present benchmarks of a single-cell proteomics data analysis strategy, referred to as DIA-ME, and explore its utility in assessing the impact of IFN-gamma treatment on the proteome.

The primary concern with the study is that the use of MBR to increase proteome coverage in single-cell analysis is not a new concept (PMID: 35810282).

Thank you for raising this point, allowing us to clarify the innovative aspects of our work. We have started working on the concept of matching enhancers more than two years ago, when we explored possibilities to increase proteome coverage in low-input data, and presented our progress on several occasions to the community. Primarily, the list of events includes the single-cell proteomics conference Vienna in September 2022, where we presented DIA-ME for the first time on stage and in a live stream. As a result, representatives of renowned MS companies contacted us and the main author was invited to two Bruker community events in November 2022 and March 2023, where he presented this work in front of other timsTOF users and to employees of Bruker Daltonics. Moreover, the corresponding author introduced the DIA-ME concept on stage and in live stream at the single-cell proteomics conference Boston in June 2023. Finally, we presented this work at the ASMS and HUPO 2023 meetings, where Thermo Fisher even used the term “DIA-ME” in their single-cell session to show the capabilities of their new Orbitrap Astral instrument. Given that we reported our process transparently over a span of 1.5 years, and since DIA-ME can be easily implemented in existing proteomics pipelines, it is not surprising that vendors and other researchers have picked up this idea in the meantime. Here, we are firmly convinced that our (early communicated) extensive evaluation has created a basis of trust that enabled this concept to spread rapidly in the community.

Nevertheless, and most importantly, all outlets that used or referred to the idea of DIA-ME, including the paper mentioned by the reviewer, only used it as a “trick” to increase proteome coverage in low-input samples, without investigating why and how it works. Our contribution now takes this several steps further, where **the innovation resides in the deep investigation of fundamental performance metrics and best practices of the concept**. Specifically, we explain the operating principle of DIA-ME, demonstrate the advantage of this concept over conventional analysis, determine the optimal size of matching enhancers and assess the reliability of the additionally obtained identifications in a systematic manner. The latter included the evaluation of false-identifications, false transfer rates and quantitative accuracy by using a mixed-species model across widely different input amounts. In addition, we determined that not all software tools handle the DIA-ME concept equally well. Specifically, we discuss the limitations of Spectronaut for low-input proteomics, and make suggestions for possible adaptations in the data filtering when employing DIA-ME in this software. Furthermore, we demonstrate that quantification of low-input data is inaccurate in DIA-NN if relying on its output matrices, and we elaborate that re-quantification by the directLFQ package solves this issue. We like to emphasize that we have experimentally verified this finding by demonstrating that additionally identified proteins contribute valuable biological information, making results of DIA-ME a closer representation of a higher input injection. Finally, we highlight the smooth integration of DIA-ME for the proteomic analysis of single-cells, where it aided to gain new insights in cellular sub-populations and protein correlation modules.

In summary, **our work is the first to broadly investigate performance metrics of DIA-ME**, and we provide guidelines with regard to experimental set-up and data analysis. This should provide a solid foundation for adoption and proper use of DIA-ME in the field.

To clarify the aspect of novelty in the manuscript, we have adapted the introduction to indicate that previous studies have empirically shown that inclusion of a higher input sample leads to better data coverage, while stipulating that our study formalizes the conceptual foundation and performance characteristics of DIA-ME.

It needs to be demonstrated that DIA-ME surpasses the performance of a GPF-based library.

We assume that the reviewer refers to gas-phase-fractionation (GPF) performing consecutive analysis of samples with a narrowed MS1 range to facilitate isolation of low-abundance signals for library generation. GPFs were recently used for single-cell analysis by serving as a matching resource for WISH-DIA data (PMID: 37737208), similar to our approach, in which higher input samples are added to the search. Specifically, the authors used six GPFs to overcome the highly convoluted spectra that resulted from their wide isolation windows in this type of acquisition. While generation of GPFs is rather straightforward on Orbitrap instruments, it requires tailored window design for diaPASEF acquisition, as was used in our study, making it challenging to establish such an approach. Therefore, we consulted the work of Penny J. *et al.* (PMID: 36876969), who created multiple individual schemes that collectively resembled the m/z-to-ion mobility space of a regular diaPASEF acquisition, but who had minor success expanding the proteome coverage when using this as a library in comparison to library-free analysis. Nevertheless, **we performed a new experiment to compare DIA-ME to GPFs** using high-input samples as a matching resource. To this end, we created the fractions by dividing our initial diaPASEF method (please refer Fig. S1A) into six individual acquisitions (panel A below) and tested it against DIA-ME, in which low-input and ME runs are recorded using an identical acquisition scheme (according to Fig. S1A). Hence, we acquired MEs and GPFs from 10-ng samples in triplicates and used them to supplement the library-free analysis of a series of 1-ng samples (16 replicates) in DIA-NN. In contrast to the analysis without matching resource (“MBR”), both DIA-ME approaches considerably improved the number of protein identifications, while the addition of MEs or GPFs resulted in equally high protein numbers (panel B below).

Although GPFs comprise more peptides due to the fractionation, i.e. provide a broader matching space, they did not lead to more identifications, similar to the saturation that we observed for higher input MEs before (Fig. 1C in the revised manuscript). In turn, GPFs entail multiple runs per replicate, letting instrument time scale with the number of fractions that are used in comparison to MEs (six fractions x three replicates = 18 injections in our experiment), while constructing the window scheme requires intensive manual operations (or coding expertise). In contrast, generation of MEs is effortless, does not require extensive instrument time (3 injections in this experiment) and they can even serve as a bulk proteome in the analysis, which can be compared to single-cell injections, whereas the results from GPFs cannot be easily merged.

In conclusion, DIA-ME achieves the same performance with significantly less effort. Although this is an interesting observation, we suggest not including this figure and its explanation in the manuscript, since the analysis does not fit well with its main focus, and since the data density is already very high. Instead, this information will be available in this rebuttal, which will be published with the paper when it is (hopefully) accepted for publication.

The validation of DIA-ME by spiked sample co-analysis was performed using 1 ng injections. Can the authors demonstrate that their method is also reliable when applied to single cells?

The selection of 1-ng samples was a conscious decision that allowed accurate assessment of false-identifications as the number of erroneously transferred *E.coli* peptides was already close to zero at 1-ng input amount (around 20 – 70 peptides per sample in DIA-NN depending on the analysis), while it still represents an atypically low quantity approaching that of a single cell. We therefore anticipated that decreasing the input would lower the number of peptides even further, hampering proper calculation of false identifications.

Nevertheless, we tried to address potential concerns about the DIA-ME approach at single-cell level by providing two new analyses:

First, we wanted to exclude that DIA-ME leads to the unspecific identification of proteins even in the absence of a corresponding signal. We therefore analyzed empty samples without cells (FACS droplets and plain buffer), which we prepared the same way as we did for single cells (panel A below; **added to the revised manuscript as new Fig. S9C**). Reassuringly, MBR and DIA-ME analysis both resulted in the identification of zero proteins, showing that proteins in our experiment did not originate from contaminations and that peptide transfer from MEs does not happen randomly. Likewise, we did not identify proteins in LC-blank-runs, each measured after a single-cell run, testifying that carryover effects did not compromise our results.

Second, and also in response to the suggestion of reviewer #3, we performed an additional experiment, in which we investigated false identifications by co-analyzing our single cell data with spiked MEs from the *E.coli* experiment of the manuscript (panel B below). We observed generally higher false-positive rates (FPR) on peptide-level compared to our results from 1-ng samples (cf. Fig. 2A in the revised manuscript). However, this can be attributed to the reduced presence of detectable peptides and their lower abundance in single-cell data since FPR was already increased for the analysis without matching (“Indiv”), indicating higher error rate in the first search. Yet, it shows the generally higher error susceptibility in the

analysis of single-cell data, which strongly emphasizes the use of adequately evaluated software tools and enforced data filtering, as provided in this work. When we employed matching for single-cell analysis, it resulted in slightly higher FPR, which we similarly observed for the entrapment in our previous analyses (see Fig. 2A and S4A in the revised manuscript). Importantly, we could not determine FPR for the analysis between single-cell inputs, which would have given an indication for the expected rates without DIA-ME. Still, since we observed equivalent FPR between DIA-ME searches and the analysis of equal inputs before (“1:1”, Fig. 2A and S4A in the revised manuscript), we conclude that there is no elevation in FPR due to co-analysis with MEs samples, still showing ratios of less than 1%.

New Fig. S9C

Consequently, these data show that the DIA-ME approach identifies low-abundance signals with high specificity even at single cell level, as it does not increase false identifications to a substandard level, if at all in comparison to searches without MEs.

The scalability of DIA-ME depending on the number of single cells analyzed needs to be investigated.

We followed the reviewer’s advice and performed an analysis of different numbers of single cells (ranging from 18 to 143) that were searched with a fixed number of 10-cell MEs to investigate the scalability of our approach. Starting with 18 individual cells, we doubled the number of analyzed cells in each search and assessed whether groups of cells (colored accordingly) showed differences in identified proteins across searches (figure below, added to the manuscript as new Fig. S9E). As we anticipated in the manuscript, the number of protein identifications per cell remained constant irrespective of the number of cells that were involved in the analysis (e.g. all 18 cells in dark blue showed equivalent results when analyzed alone or together with light blue, light red and red cells). Consequently, the effect of matching signals to MEs is not “used up” and each cell benefitted equally from DIA-ME. Hence, our findings justify the statement that DIA-ME is a highly scalable approach that only requires a few MEs for the analysis of a theoretically limitless number of single-cell samples. To support this notion, we have added this figure as Fig. S9A to the revised manuscript.

Do the extra IFN-gamma responders detected with DIA-ME using 200 pg injections show the same changes in the 200 ng experiment?

Yes, the majority of additionally identified proteins with DIA-ME show similar changes in the 200-ng and 200-pg experiments. Specifically, MTOR, TAP2 and HLA-A/B are similarly upregulated over the treatment period in both datasets, while they have not been identified by “MBR” alone, as shown in Fig. 5D/E and S7F/G. To address this question further in the manuscript, we performed an overrepresentation analysis of up-regulated proteins from 200-pg analysis with and without MEs and compared the results to the 200-ng experiment (please refer Fig. 5F and S7H). Not only did we find a strong overlap between the enriched terms from both input amounts (bold terms), but DIA-ME analysis also led to higher coverage and enrichments of these terms, while even indicating several terms that were not covered by conventional MBR analysis.

To allow the tracking of specific proteins across multiple panels, we **changed the appearance of Fig. 5A and S9C**, so that dots of known IFN- γ -responding proteins now also show their derivation from conventional or DIA-ME analysis.

I could not find any information on how multiple testing correction was performed.

We have now added Benjamini-Hochberg-corrected p-values to the supplementary tables.

Given the use of MBR, the traditional entrapment FDR calculation is no longer applicable. This explains the high values noted for Spectronaut. Error control should therefore be performed only using the FPR and FTR.

The reviewer brings up a fair point. We therefore removed Fig. S4A from the manuscript as it showed the results of a simplified FDR calculation. Instead, **we re-analyzed our data** to now provide precursor-level

FPR and FTR since they are a closer reflection of the identification process, presented in a new set of figures (new Fig. 2A and C, shown below).

New Fig. 2A

New Fig. 2C

Moreover, in the protein-level analyses (original Fig. 3A and C), we tightened the default protein q-value cutoff (run-wise) in Spectronaut from 5% to 1% to improve comparability to DIA-NN, by which we intend to account for the high identification load of ME runs that inflates the global q-value. **The revised figures are now included in the supplement as Fig. S4A and S4C (also shown below).**

As we changed the protein-level q-value settings in Spectronaut, we found that FPR was significantly reduced for DIA-ME analysis, showing values of only up to 1.5% (panel A below; 2.1% with default filtering). Yet, we observed a slight increase in FTR in cases where equal injection amounts were co-analyzed (1:1) (panel B below), showing that by this filter more human proteins were affected among the transferred signals. Remarkably, this was also the case when 100x ME samples were added to the analysis, where the majority of the transferred identifications were false-positives (e.g. up to 75% in case of the 20% *E. coli* sample, panel B below). This is an important finding, since on the one hand these results suggest that the DIA-ME analysis can be applied using Spectronaut, however, it needs to be carefully limited to 10x MEs and requires adapting the run-specific protein q-value cutoff to ensure solid control of false-identifications. In turn, we found consistently lower FPR and less erroneous matching in DIA-NN keeping default settings, which justifies its application in this work. We have added a statement in the manuscript to raise awareness that Spectronaut and DIA-NN perform differently in the DIA-ME approach.

Previous Fig. 3A updated and moved to supplement Fig. S4A

Previous Fig. 3C updated and moved to supplement Fig. S4C

I recommend that the authors include a supplementary table detailing the full configurations of the software used in their analysis.

We agree that detailing the exact software configurations and compositions of raw files for each analysis is relevant to make this work easier to comprehend. Consequently, **we added the respective table to the supplementary material (Data S1)**. As a commentary note: we kept default settings in Spectronaut (changes in q-values were performed externally) and we only made slight adaptations to the searches in DIA-NN that we indicated in the method section of the manuscript (please refer to Material & Methods “Raw data processing”).

Why are intensity sums the same between 2 and 5 ng on Figure S1C?

We also wondered while plotting, and we suppose it is a result of chromatographic error, but the data still serves the purpose of showing a wide range of *E.coli* identifications.

Reviewer #2

The manuscript by Krull and coauthors presented a detailed evaluation of the MBR feature of DIA-NN and Spectronaut software for low-input proteomics, with preliminary data for single-cell proteomics. While many evaluations have been done for bulk-scale DIA analysis, it has great value to make such a study focus on low-input datasets, given the growing interest in single-cell analysis nowadays. The authors used two-species proteomes to study the quantitative precision, as well as false discovery rates at different input levels. The optimized workflow was used to study interferon-gamma treatment of tumor cells. While the data present here is a valuable resource for the field, the overall innovation is low. The reviewer doesn't believe it should be published in NC.

We thank the reviewer for recognizing the value of our work, and we are happy to clarify its innovative nature in our responses below.

The use of high-input DIA samples analyzed together with low-input samples was widely used in the field and widely appeared in the vendor's technical report. The major contribution of this manuscript is the optimization of sample sizes for high-input samples. They also suggest normalizing the data using the directLFQ package to improve the precision. However, overall such improvement is incremental.

Thank you for raising this point, allowing us to clarify the innovative aspects of our work. We have started working on the concept of matching enhancers more than two years ago, when we explored possibilities to increase proteome coverage in low-input data, and presented our progress on several occasions to the community. Primarily, the list of events includes the single-cell proteomics conference Vienna in September 2022, where we presented DIA-ME for the first time on stage and in a live stream. As a result, representatives of renowned MS companies contacted us, and the main author was invited to two Bruker community events in November 2022 and March 2023, where he presented this work in front of other timsTOF users and to employees of Bruker Daltonics. Moreover, the corresponding author introduced the DIA-ME concept on stage and in live stream at the single-cell proteomics conference Boston in June 2023. Finally, we presented this work at the ASMS and HUPO 2023 meetings, where Thermo Fisher even used the term “DIA-ME” in their single-cell session to show the capabilities of their new Orbitrap Astral instrument. Given that we reported our process transparently over a span of 1.5 years, and since DIA-ME can be easily implemented in existing proteomics pipelines, it is not surprising that vendors and other researchers have picked up this idea in the meantime. Here, we are firmly convinced that our (early communicated) extensive evaluation has created a basis of trust that enabled this concept to spread rapidly in the community.

Nevertheless, and most importantly, all outlets that used or referred to the idea of DIA-ME, including the paper mentioned by the reviewer, only used it as a “trick” to increase proteome coverage in low-input samples, without investigating why and how it works. Our contribution now takes this several steps further, where **the innovation resides in the deep investigation of fundamental performance metrics and best practices of the concept**, where the reviewer mentions only some of many aspects that we addressed. Specifically, we explain the operating principle of DIA-ME, demonstrate the advantage of this concept over conventional analysis, determine the optimal size of matching enhancers and assess the reliability of the additionally obtained identifications in a systematic manner. The latter included the evaluation of false-identifications, false transfer rates and quantitative accuracy by using a mixed-species

model across widely different input amounts. In addition, we determined that not all software tools handle the DIA-ME concept equally well. Specifically, we discuss the limitations of Spectronaut for low-input proteomics, and make suggestions for possible adaptations in the data filtering when employing DIA-ME in this software. Furthermore, we demonstrate that quantification of low-input data is inaccurate in DIA-NN if relying on its output matrices, and we elaborate that re-quantification by the directLFQ package solves this issue. We like to emphasize that we have experimentally verified this finding by demonstrating that additionally identified proteins contribute valuable biological information, making results of DIA-ME a closer representation of a higher input injection. Finally, we highlight the smooth integration of DIA-ME for the proteomic analysis of single-cells, where it aided to gain new insights in cellular sub-populations and protein correlation modules.

To clarify the aspect of novelty in the manuscript, we have adapted the introduction to indicate that previous studies have empirically shown that inclusion of a higher input sample leads to better data coverage, while stipulating that our study formalizes the conceptual foundation and performance characteristics of this approach.

There is no new software or new algorithms were developed in this work. The DIA-ME essentially is a data analysis workflow, which doesn't justify being highly innovative.

DIA-ME is an experimental approach to improve proteome coverage in low-input samples by label-free proteomics. Indeed, it makes use of existing software, however we emphasize specific settings to ensure proper performance of DIA-ME in these respective tools. Improving proteome coverage and data completeness is an unmet need in single-cell proteomics, and therefore we believe that DIA-ME provides a valuable and readily accessible workflow to the field. The innovation of our work resides in the fact that it is the **first to broadly investigate performance metrics of DIA-ME, and we provide guidelines with regard to experimental set-up and data analysis**. This should provide a solid foundation for adoption and proper use of DIA-ME in the field. Please see our response to the previous question where we highlighted specific achievements of our work in more detail.

Figure 5F, the correlation map indicates the correlation between 0 hours and 24 hours is higher than 18 hours. More surprisingly, at each time point, only 1-2 replicates showed low correlations with 0 hours, and there are sub-clusters in each time point, which are very confusing. The authors should investigate why the cell response is so diverse at each time point, giving these are bulk-scale (diluted) experiments.

We thank the reviewer for this comment. As a first clarification: in the IFN- γ experiment, replicates originate from three different cultures (biological replicates) per time-point, which explains the presence of inherent differences within time-points (please refer method section “Cultivation of U-2 OS cells and Interferon-gamma treatment”). Moreover, please note that the correlation scale in Fig. 5F (renumbered as Fig. 4F in the revised manuscript) is very high (> 0.88), while also being very narrow ($0.88 - 0.99$), making replicate discrepancies appear more pronounced. To answer why correlations between 0 and 24 hours are higher than to 18 hours, we want to point out that the treatment effect on the proteome has peaked at 18 hours, which can also be derived from PCA analysis in Fig. 5G (now Fig. 4G). Although we noticed that some IFN- γ responsive proteins reached their highest differential regulation at 18 hours, many of the

hallmark proteins continue to rise until 24 hours (Fig. 5E), indicating that the IFN- γ treatment performed as expected.

Prompted by the reviewer's question, we also revisited the data of the IFN- γ treatment of the high-input sample (Fig. S7). When doing so, we became aware of a mistake in the code the generated panels S7B and S7D. After correcting this, the heatmap (Fig. S7B) now also indicated the highest disparity between 0 and 18 hours in bulk analysis, while the PCA analysis clearly separated the different time points (Fig. S7D). Thereby, this reconciles the results from the high- and low-input experiments.

The updated figures S7B and S7D have been included in the revised manuscript.

The data quality of single-cell is very poor. There is no blank controls for this experiment (blanks mean empty FACS droplet with all sample prep procedures). The author should include such blanks to demonstrate the data is actually from single cells, as many recent data shows that background contamination is a major problem of single-cell proteomics.

We agree that background contamination constitutes a potential risk in single-cell proteomics, given that the analyzed material is of very low abundance. To bolster confidence in our results, **we assessed our protocols by an additional experiment**, in which we sorted and prepared empty FACS droplets, i.e. that did not contain cells. Reassuringly, analysis of these samples with and without 10-cell MEs lead to the identification of zero protein groups (new Fig. S9C below), testifying the absence of contaminants. Moreover, we want to highlight that we intensively prepare our LC systems prior to single-cell analysis to ensure that there is no carryover between the runs. Indeed, we identified zero proteins in blank runs that we routinely acquired between single-cell injections in our IFN- γ experiment (panel below) using the same gradient and acquisition method.

We added this figure to the manuscript as new Fig. S9C.

New Fig. S9C

Figure 7A. The improvement of DIA-ME is relatively moderate for single-cell data.

Given that the dynamic range of proteins in a cell follows a sigmoidal function, a substantial improvement in sensitivity is required to identify proteins beyond the most abundant ones. Therefore, we adjudge the observed increase of around +16% proteins in our experiment as a significant enhancement in comparison to conventional analysis. In this sense, we also believe that the effect of DIA-ME benefits from an increased proteomic depth, and will be even more pronounced with optimized sample preparation protocols and with more sensitive mass spectrometers, as we pointed out in our discussion. However, the effect of DIA-ME is not only limited to the detection of more proteins, but is also evident by a higher peptide coverage per protein (please refer Fig. 6D) and higher data completeness (please refer Fig. 1D).

Considering the reviewer’s comment, **we revised the representation of our data** and substituted our previous Fig. 7C, which only adds minor value to the results, with a histogram showing the extension of the dynamic range by the DIA-ME approach (**new Fig. 6B**). In addition, **we updated Fig. S9B**, which now also shows the increase of unique proteins compared to the conventional analysis.

New Fig. 6B

Updated Fig. S9B

PCA analysis indicates the two cell state only resolved at PC2 space, which indicates the quantitative precision is low for the overall workflow.

Indeed, cells from control and IFN- γ -treated condition resolved along PC2 since this component accounts for the second highest variance in the data set (previous Fig. 7F). The reviewer’s comment **prompted us to revisit our data** to find an explanation for the bigger effect size in PC1 space. Triggered by the cluster analysis in Fig. 6F (previous Fig. 7H) we discovered that PC1 can be explained by differences in the expression of proteins involved in oxidative stress response, such as GAPDH, CAT and ARG1 (figure below). This effect appears to be independent of the treatment IFN- γ , and may result from different oxygenation during cell culture. We thank the reviewer making us aware of this gap in our explanation, to show that **PC1 has a biological origin instead of indicating low quantitative precision. We have rearranged our previous Fig. 7F to accommodate the updated figure below (now Fig. 6E) to highlight the respective proteins responsible for separation along PC1 and PC2.**

Previous Fig. 7F updated, now Fig. 6E

The author did two studies of the interferon-gamma treatment (Figure 5-7). Returning to the value of single-cell proteomics, did the authors identify new functional modules missed in bulk-scale (diluted) experiments? What's the biological significance of performing the single-cell assay if you can get the same or better answer from the bulk assay?

Indeed, we made several novel observations in the single cell data that remained hidden in the bulk analysis. For instance, single-cell proteomics allows uncovering transitions of cellular states and the presence of subpopulations either in IFN- γ -treated or control cells, both of which we explored in our work. Specifically, IFN- γ -treated cells showed a gradual polarization in PCA analysis (please see Fig. 6E in the revised manuscript), indicating differences in the strength of IFN- γ response between individual cells. Similarly, we observed gradual differences between control cells (Fig. 6E), collectively resulting in a continuum of cell states rather than showing two discrete populations. Moreover, using cluster analysis, we identified a group of cells that experienced oxidative stress (Fig. 6F), which was not apparent in bulk analysis (cf. Fig. 6B and S7F). In addition, we learned that these oxidative stress proteins show gradual differences in expression. One notable example is GAPDH, usually considered as a “housekeeping” protein, which differed up to 8-fold in expression between individual cells (Fig 6E). Furthermore, we performed correlation analysis between protein pairs within the same cell to characterize cellular

heterogeneity and transition between cellular states even in more detail (Fig. 7 and S10). As we explored protein dependencies, we discovered several interesting examples of proteins that are involved in similar processes, but that showed mutually exclusive expression patterns (Fig. S10B). Specifically, we encountered the “redox-switch” of GAPDH upon oxidative stress (PMID: 37024754) using network analysis (Fig. 7C), which causes the redirection of the glycolytic flux into the pentose-phosphate pathway, while resulting in a down-regulation of TCA cycle activity and oxidative phosphorylation (Fig. 7D). Hence, **our work shows that we can retrieve biological significance from single-cell analysis that goes beyond traditional bulk proteomics.**

To better highlight these findings, **we rearranged Fig. 7** to fully dedicate this to the interpretation of the single-cell data.

Figure 4 B and 4C, the authors used different colors to indicate different samples in one plot. However, everything is mixed. It is very challenging to identify the data from each sample. It is suggested to separate them into different plots.

We thank the reviewer for their suggestion and agree that this type of scatter is difficult to grasp and interpret. We have **exchanged our previous Fig. 4C by a new figure** that sums the quantitative deviations of *H.sapiens* and *E.coli* identifications in the performed searches in a (hopefully) more insightful manner (new Fig. 3C, see below). These data show that both *H.sapiens* and *E. coli* proteins show the expected ratios, independent of the ME amount. We also added a small legend to Fig. 4B (not shown here, renumbered as Fig. 3B in the revised manuscript) to clarify that colored markers originate from additionally identified proteins in the respective DIA-ME analysis.

Previous Fig. 4C updated, now Fig. 3C

Figure 6F, there is some disagreement between MBR and DIA-ME. The author should discuss why some enrichments were only observed in MBR. There are several 0 and negative counts in MBR data, which should be clarified.

We apologize for some misunderstanding that may arise from how these data were displayed. Fig. 6F (renumbered as Fig. 5F in the revised manuscript) shows the overrepresentation analysis for the up-regulated proteins from MBR and DIA-ME analyses, where set sizes are illustrated as overlapping bar plot. The numbers next to the bars indicate the **additional** contribution of DIA-ME, i.e. how many **extra** proteins were present for this term in addition to the proteins already identified by MBR analysis. Most terms show positive values, indicating that DIA-ME populated those terms with more proteins, or produced terms that were solely enriched by DIA-ME (solid blue bars). A number of zero indicates that DIA-ME and MBR covered the same number of proteins for the term, while in case of the few negative numbers DIA-ME analysis did not result in a significant enrichment. Notably, the corresponding proteins in the latter cases also showed an up-regulation in DIA-ME, however, the terms did not pass the default FDR filtering of the MSigDB hallmark enrichment. Comparing terms from this 200-pg experiment with the enrichment from 200-ng bulk analysis (terms displayed in bold font), showed that the enrichment from DIA-ME analysis was a closer reflection of the higher input sample than the conventional analysis. This observation highlights the utility of our approach making biological inferences.

To aid the understandability of this graph, **we added algebraic signs to the numbers and changed their color appearance, and modified the text to better explain the interpretation of this figure.**

Updated Fig. 5F

Reviewer #3

The authors describe an improved mass spectrometry data acquisition method for single cell proteomics. In short, by providing proteomics search engines with additional data from a higher-sample input run, the identification efficiency can be improved through the search engine's 'match-between-runs' strategy. This method, coined DIA-ME (matching enhancer), is optimized and benchmarked on technical spike-in samples, and then evaluated on biological samples.

While the authors provide an extensive dataset to evaluate the effectiveness of the method, the current work lacks an in-depth analysis of the match-between-runs implementations in the used search engines. Indeed, the two closed-source tools that are used by the authors (DIA-NN and Spectronaut) behave very differently across the testcases, but the cause of these differences are not thoroughly investigated. Addressing the comments below could drastically improve the quality of the manuscript in terms of data analysis. I did not fully assess the acquisition and biological use case sections of the manuscript, as these are somewhat outside of the scope of my expertise.

We thank the reviewer for seeing the value of our work and for constructive comments to perform a more detailed assessment of the performance of DIA-NN and Spectronaut in their handling of the DIA-ME concept. As indicated in greater detail in our responses below, we have now conducted such deeper comparative analysis, which pointed to distinct differences in performance, leading us to provide guidelines in using specific (q-value) settings that should be considered for optimal operation of DIA-ME.

*The authors consistently report results at the level of protein-group. While this is useful from a biological perspective, the search engines are identifying and transferring precursor identifications. For more concrete comparisons, and to avoid any biases from steps such as protein inference, I therefore strongly recommend reporting precursor level identification rates, false discovery rates, and false transfer rates. Especially for reporting the latter, precursor-level data would allow the authors to correctly account for the amount of *E. coli* vs human peptides in each database instead of the number of proteins, which does not account for differences such as protein lengths.*

Since we see DIA-ME as an experimental approach, we only compared the software on protein level, however, we agree with the reviewer that reporting of results from precursor identifications at this point would give even clearer insights. **We therefore performed the respective analyses and added the results to the revised manuscript (Fig. 2).** These analyses showed that the determined false-positive (FPR) and false-transfer rates (FTR) were considerably lower than their respective value on protein level, since false-identifications of *E.coli* proteins were mostly based on a small number of peptides. Yet, the results complement our previous observation that FPR is significantly elevated in the context of Spectronaut (new Fig. 2A) due to its erroneous signal matching (new Fig. 2B). In contrast, DIA-NN shows reasonably low FPR and FTR for matching among equal injection amounts (1:1) and when using higher input MEs, confirming the applicability of the DIA-ME approach in this software.

New Fig. 2A

New Fig. 2C

In addition to adding these new data, we moved the results of protein-level false-positive and false-transfer rates to the supplementary material (Fig. S4A and C in the revised manuscript). On request of reviewer #1, we further removed false-discovery rates from the supplement and thus did not perform this analysis on precursor-level.

The differences between DIA-NN and Spectronaut should be further investigated. The authors allude to a less stringent filtering in its 'first search' for Spectronaut. Additionally, MBR strategies are originally meant to be used between runs for similar samples. In this case, the ME run can be regarded as very different from the low-input sample runs. Therefore, depending on how FDR filtering is applied by the search engines for the initial search (on a per-run basis, or across all runs), ill-estimated FDR thresholds could lead to either too many false identifications or a decrease in identification sensitivity. Unfortunately, DIA-NN and Spectronaut are closed-source and can therefore be regarded as black boxes. Nevertheless, both (potential) issues could be easily investigated by first searching the ME runs separately, extracting the resulting spectral libraries, and then searching the low-input samples with these spectral libraries. Applying this additional test for both search engines, might shed some light on the reported differences.

We thank the reviewer for the important question, allowing us to better highlight that DIA-NN and Spectronaut perform differently when handling DIA-ME data, which is highly relevant for the community when adopting DIA-ME.

As an initial comment, we want to point out that our work focusses on assessing the matching capabilities of Spectronaut and DIA-NN, especially in the context where low and high input samples are co-analyzed. Based on our observations, we argued that feature matching, i.e. the second pass of the search, is considerably less controlled in Spectronaut than in DIA-NN (please see previous Fig. 3A and C), while we found indications for a lenient filtering in Spectronaut in our initial version of the manuscript (see Fig. S5).

As requested by the reviewer, **we conducted a number of additional analyses** to investigate which features determine the performance of DIA-NN and Spectronaut in the context of low-input samples and for the application of DIA-ME. Since the reviewer's questions directly concern the inner workings of these software tools, we have mainly relied on the peptide-level analysis conducted in response to the previous question of this reviewer (above).

To analyze the first pass of the search, we joined a *H.sapiens* and *E.coli* database (fasta files) that served as entrapment for the library-free analysis of individual raw files ("Indiv"), and reported a false-positive rate (FPR) that was lower using Spectronaut than for DIA-NN (**new Fig. 2A in the revised manuscript**). While the architecture of DIA-NN consists of two peptide-centric search instances, Spectronaut's first search is spectrum-centric in the context of library-free analysis, which might therefore explain the very different results for searches without (entrapped) matching ("Indiv" and "MBR"). In turn, co-analysis with other spiked 1-ng samples ("1:1") led to a substantial increase in FPR in Spectronaut (new Fig. 2A), which employs a peptide-centric approach for the second pass of its search. Nevertheless, as the reviewer indicated, lenient (hidden) FDR filtering after the initial search, but also the very different library sizes (*in silico* prediction versus experimentally generated internal library) could lead to similar observations. Hence, we used spectral libraries that we pre-generated from our spiked 10-ng ME runs to resemble the internal library from the second pass of the analysis, investigating the peptide-centric searches of DIA-NN and Spectronaut (panel A below), according to the reviewer's suggestion.

To enable direct comparison, we again included the results of precursor-level FPR from our library-free analysis that was described in our previous statement (**new Fig. 2A**).

New Fig. 2A (see previous statement)

Interestingly, analyzing 1-ng samples in spiked library-based searches without matching (“Indiv”) only resulted in an FPR of 0.1% for DIA-NN (panel A above), which was approximately half of the previous entrapped library-free approach (around 0.2%, panel B above), while showing substantially higher FPR of around 0.17% for Spectronaut (panel A; around 0.06% in panel B). This observation suggests that DIA-NN potentially benefitted from the significantly smaller query space in comparison to the *in silico*-predicted library, as was described for DIA analysis (PMID: 28825704). In contrast, FDR-control of library-based (peptide-centric) analysis in Spectronaut seemed to be significantly worse than in library-free (spectrum-centric) analysis, despite the smaller library size. Since the second pass of the search can be regarded as library-based analysis, this observation underpins our previous results that matching is more error-prone in Spectronaut. To highlight this point further, we co-analyzed our 1-ng samples with spiked samples of equal input amount using a 10%-spiked library and enabled matching (“1:1”, panel A), which constitutes two consecutive library-based analyses. The resulting FPR in Spectronaut exceeded the rate that we observed when the initial search was library-free, showing a value of more than 0.4% (0.25 – 0.4% in panel B). This observation is yet another indication that Spectronaut’s peptide-centric matching shows compromised FDR control in comparison to its initial spectrum-centric search in library-free analysis. Reassuringly, the same analysis in DIA-NN resulted in a FPR of only around 0.125% (panel A) which was considerably lower than with the *in silico*-generated library in the first search (0.175 – 0.25% in panel B).

Prompted by the reviewer’s comment about FDR filtering, we **additionally performed an analysis** of the utilized q-values in Spectronaut and DIA-NN by receiver operating characteristics (ROC) to further investigate the differences between these software tools that have led to our afore discussed observations. We found that the global protein q-value in Spectronaut (“PG.Qvalue”) is seriously affected by the presence of MEs due to their high identification load, while this does not seem to be the case in the context of DIA-NN (“Lib.PG.Q.Value”) (**new Fig. 2B below**). Hence, we decided to tighten the run-wise q-value filter for proteins in Spectronaut from 5% (default) to 1% for our DIA-ME approach in the revised manuscript to improve comparability to DIA-NN on protein-level. Notably, however, all (even run-specific) q-values were less FDR controlled using DIA-ME in Spectronaut (new Fig. 2B), which calls its score calculation into question and potentially explains the excessive presence of false identifications using default filtering for the analysis of low-input data in this software. Investigating filters in DIA-NN, we found that its run-specific protein q-value (unused in default settings) offers great specificity to discriminate false identifications and therefore constitutes a useful addition for the filtering of low-input data in this software (**new Fig. S4B**).

New Fig. 2B
(default q-value filters)

New Fig. S4B
(unused q-value filters in DIA-NN)

As we changed the protein-level q-value settings in Spectronaut, we found that FPR was significantly reduced for DIA-ME analysis, showing values of only up to 1.5% (panel A below; 2.1% with default filtering). Yet, we observed a slight increase in FTR in cases where equal injection amounts were co-analyzed (1:1) (panel B below), showing that by this filter more human proteins were affected among the transferred signals. Remarkably, that was also the case when 100x ME samples were added to the analysis, where the majority of the transferred identifications were false-positives (e.g. up to 75% in case of the 20% *E.coli* sample, panel B below). This is an important finding, since on the one hand these results suggest that the DIA-ME analysis can be applied using Spectronaut, however, it needs to be carefully limited to 10x MEs and requires adapting the run-specific protein q-value cutoff to ensure solid control of false-identifications. In turn, we found consistently lower FPR and less erroneous matching in DIA-NN keeping default settings, which justifies its application in this work. We have added a statement in the manuscript to raise awareness that Spectronaut and DIA-NN perform differently in the DIA-ME approach.

In conclusion, we have conducted a more thorough comparison of DIA-NN and Spectronaut pointing to salient differences between these software tools in a DIA-ME approach. The displayed figures in this response and a detailed description of these data have now been included in the revised manuscript.

The authors benchmark the method on low-input samples instead of actual single cells. The reported numbers of identified protein groups are significantly lower for the actual single cell experiments (~500) than on these low-input samples (~2500). To address potential concerns on the transferability of the low-input sample benchmarks to actual single cell experiments, the authors could relatively easily assess the false transfer rates by searching the single cell runs with the spike-in ME runs.

We appreciate the reviewer's comment and performed the suggested analysis. Specifically, we have searched our single-cell runs with human and *E.coli* fasta files and in conjunction with the spiked 1-ng or 5-ng ME samples that we generated in the context of our two-proteome model experiment.

We observed generally higher false-positive rates (FPR) on peptide-level compared to our results from 1-ng samples (cf. Fig. 2A in the revised manuscript). However, this can be attributed to the reduced presence of detectable peptides and their lower abundance in single-cell data since FPR was already increased for the analysis without matching ("Indiv"), indicating higher error rate in the first search. Yet, it shows the generally higher error susceptibility in the analysis of single-cell data, which strongly emphasizes the use of adequately evaluated software tools and enforced data filtering, as provided in this work. When we employed matching for single-cell analysis, it resulted in slightly higher FPR, which we similarly observed for the entrapment in our previous analyses, while showing equivalent FPR between DIA-ME searches and the analysis of equal input amounts (see Fig. 2A and S4A in the revised manuscript). Importantly, however, we could not determine FPR for the respective analysis between single-cell inputs ("1:1"), which would have given an indication for the expected rates without DIA-ME. Concomitantly, we found higher false-transfer rates (FTR) for matching in single-cell data (panel B below), however, based on our previous

observation that FTR was lower in DIA-ME compared to analysis of equal inputs (Fig. 2C), we conclude that is not a result of co-analyzing higher input samples. Consequently, **these data show that the DIA-ME concept can be applied at single cell level**, as it does not increase false-identifications to a substandard level (still below 1%), if at all in comparison to searches without MEs.

From communications with colleagues, it seems that the acquisition of a bulk sample along single cell samples for use with DIA search engine MBR methods is already widely applied. It therefore seems that the authors have somewhat overstated the novelty of this approach in the manuscript. I would therefore recommend positioning the manuscript as an optimization and validation of the approach rather than an introduction of a novel method.

We thank the reviewer for the comment. In response to similar comments by the other reviewers, we explained that we have been very active in disseminating our findings at various conferences and MS user meetings to indicate the power of DIA-ME. Given that DIA-ME can be easily integrated in existing workflows, we believe that this has contributed to rapid adoption in the field. Nevertheless, to clarify the aspect of novelty in the manuscript, **we have adapted the text in the introduction** to indicate that previous studies have empirically shown that inclusion of a higher input sample leads to better data coverage, while stipulating that our study formalizes the conceptual foundation and performance characteristics of DIA-ME.

All scripts and source code for the work's data analysis seems to be missing from the manuscript submission. Sharing the Jupyter notebooks (for Python code) and R scripts could improve the clarity of the applied methods (e.g. calculation of the false transfer rate) and is a must for reproducibility of the work. Secondly, the raw proteomics data could prove to be a highly valuable resource to improve DIA-ME data analysis methods. Therefore, uploading SDRF metadata files alongside the raw data would strongly facilitate further development by the community. Thirdly, the authors seem to have only uploaded mzML data to PRIDE. For full reproducibility of the work, the unprocessed raw mass spectrometry files (.raw or .d) should be uploaded.

We apologize for not having shared our pipelines earlier and we have now uploaded all relevant code to our GitHub page (<https://github.com/krijgsveld-lab/DIA-ME>) to make it freely accessible.

Moreover, we created an SDRF metadata file for our mixed-species model experiment to facilitate interpretability of our analysis and we uploaded it together with the respective raw data to PRIDE. We also re-uploaded all timsTOF raw data as unprocessed files and we indicated the accession numbers in the manuscript, respectively (Two-proteome model experiment: PXD053462; 200-ng IFN- γ treatment experiment: PXD048162; 200-pg IFN- γ treatment experiment: PXD053473; Single-cell experiment: PXD053464).

The manuscript contains many figures of which some only add minor value. I would suggest somewhat limiting the figures in the main manuscript to enhance readability, interpretability and minimize redundancies.

We agree with the reviewer's comment that some figure panels were redundant or only added incremental insight. To improve this and maximize information content, and as indicated throughout this rebuttal, we have therefore replaced several figure panels with updated versions, we have added new figure panels resulting from additional analyses based on reviewers' comments, and we have rearranged several figure panels between main and supplemental figures.

Reviewers' Comments:

Reviewer #1:

Remarks to the Author:

I agree with the other reviewers that DIA-ME lacks novelty as a method. However, the authors have successfully addressed all suggestions. In my view, this brings the manuscript to a level comparable to several similar works recently published in Nature Communications. Therefore, I recommend publication.

- Comparing S7F to HF5E, HLA-B is shown instead of STAT1.

Reviewer #2:

Remarks to the Author:

In the response letter as well as the revised manuscript, the authors made a great effort to explain the origin of the method and the innovation aspect of this manuscript. They identified the main innovation is the "deep investigation of fundamental performance metrics and best practices of the concept", as well as "broadly investigate performance metrics of DIA-ME".

This is exactly the concern of reviewer 1 and reviewer 2. While the reviewers appreciated these efforts and the value of the study, the current manuscript really doesn't add too much to the current single-cell proteomics field, as such practice is already used in many labs and vendors. The reviewer would agree it may be appropriate to publish on Nature Communications if the manuscript was submitted in 2022. However, it is 2024 nowadays, and there are many new directions/instrumentations/methods in the fields

As an optimization study, it should be more appropriate to publish in more specialized proteomics journals such as Molecular & Cellular Proteomics or Journal of Proteome Research.

In new Figure 9c, the author claims to identify zero proteins from FACS blanks, which is surprising as it is extremely difficult to get to zero proteins in ultrasensitive proteomics. Could you identify trypsin? keratins? histones? actin? Or other common contamination proteins?

In the new Figure 6E, the author explains the PC1 variation reflected oxidative stress response. The author also claims that such an observation represents the unique value of single-cell proteomics, in comparison with bulk proteomics. However, it is surprising

to see the large difference in oxidative stress, given that all the cells were cultured in the same dish. As GAPDH is the housekeeping protein and it is easy to do FACS assay, a new experiment should be done to validate this conclusion.

Reviewer #3:

Remarks to the Author:

The additional analysis and clear discussions of the newly added results are well appreciated. I can now recommend the manuscript for publication. The authors addressed all my main comments and concerns:

The authors performed a deeper comparative analysis of DIA-NN and Spectronaut, resulting in a better understanding of the differences between the two search engines' MBR approaches. Moreover, the authors now provide new and improved guidelines for configuring both search engines for DIA-ME, enhancing the value of the manuscript. They also performed the suggested evaluation of false transfer rates (FTR) with single cells, confirming the applicability of DIA-ME, albeit with slightly increased FTR. The FTR results are now appropriately shown at the precursor level instead of the protein level. All relevant code has been uploaded to GitHub, SDRF metadata files have been created, and the raw mass spectrometry data has been uploaded to PRIDE, greatly improving the reproducibility and value of the work for the community. Furthermore, the introduction was revised to position the manuscript as a formalization and evaluation of the technique.

Rebuttal II

Reviewer #1

I agree with the other reviewers that DIA-ME lacks novelty as a method. However, the authors have successfully addressed all suggestions. In my view, this brings the manuscript to a level comparable to several similar works recently published in Nature Communications. Therefore, I recommend publication.

- Comparing S7F to HF5E, HLA-B is shown instead of STAT1.

We thank the reviewer for the comment and appreciate their recommendation for publication. Regarding our Figure S7F, we want to note that STAT1 is instead shown in the adjacent Fig. S7G to allow comparison to JAK1, highlighting their activation upon IFN- γ . Likewise, STAT1 is shown in Fig. 5E since it also appeared in the cluster in Fig. 5B, while the up-regulation of HLA-B is shown in Fig. 5D.

Reviewer #2

In the response letter as well as the revised manuscript, the authors made a great effort to explain the origin of the method and the innovation aspect of this manuscript. They identified the main innovation is the "deep investigation of fundamental performance metrics and best practices of the concept", as well as "broadly investigate performance metrics of DIA-ME".

This is exactly the concern of reviewer 1 and reviewer 2. While the reviewers appreciated these efforts and the value of the study, the current manuscript really doesn't add too much to the current single-cell proteomics field, as such practice is already used in many labs and vendors. The reviewer would agree it may be appropriate to publish on Nature Communications if the manuscript was submitted in 2022. However, it is 2024 nowadays, and there are many new directions/instrumentations/methods in the fields

As an optimization study, it should be more appropriate to publish in more specialized proteomics journals such as Molecular & Cellular Proteomics or Journal of Proteome Research.

In new Figure 9c, the author claims to identify zero proteins from FACS blanks, which is surprising as it is extremely difficult to get to zero proteins in ultrasensitive proteomics. Could you identify trypsin? keratins? histones? actin? Or other common contamination proteins?

We kindly disagree with the statement that it is surprising that we are not identifying proteins in our empty samples, since we intensively prepare our LC system before single-cell analysis by de-saturating our columns and making significant effort to exclude contaminations during sample preparation. This includes among others the exclusive employment of single-use item, minimized reaction steps and the avoidance of any sort of binding material before the analytical column. In our data for the new Fig. 9C, we have therefore indeed only identified trypsin peptides in some of the measurements when we analyzed the data together with a contaminants fasta file. Analyzing the empty samples as we did with the single cells just with a human proteome database, we identified zero proteins in all runs, as reported. For further reference, we had already uploaded the respective raw data together with our single-cell runs to PRIDE.

In the new Figure 6E, the author explains the PC1 variation reflected oxidative stress response. The author also claims that such an observation represents the unique value of single-cell proteomics, in comparison with bulk proteomics. However, it is surprising to see the large difference in oxidative stress, given that all the cells were cultured in the same dish. As GAPDH is the housekeeping protein and it is easy to do FACS assay, a new experiment should be done to validate this conclusion.

First, the colloquial term “housekeeping protein” refers to the notion that its expression does not change under the influence of a perturbation of interest, however, **it does not mean that it cannot change under any circumstance**. In fact, GAPDH is a multi-functional protein that is subject to extensive regulation (PMID 25859407 and PMID 37024754 (reference #56 in the updated manuscript)). In our experiments, single cells differ in metabolic state, as evidenced by the graded expression of proteins that act in the same functional pathway along with GAPDH (i.e. oxidative stress). Therefore, concordant differences in expression of these proteins provides strong support that our observation for GAPDH is not an artifact. As an explanation for this to occur in the same cell culture well, in the manuscript we mention that this can be due to detachment of (some) cells, which is known to elevate endogenous oxidant levels (PMID 19693011 and PMID 27049945). Alternatively, this may be due to some cells being located near the surface (highly oxygenated), while others are closer to the bottom of the well (less oxygenated). Since it can be assumed that these environments are fairly similar between culture vessels using equivalent culturing conditions, the distribution of oxidant levels should be proportionally the same. Thus, this effect would go unnoticed in a bulk analysis, while being revealed if looking at cells individually.

Second, it is well-established that GAPDH drastically differs in expression between cell types, which has been shown between tissue types, both on transcript (PMID 15769908 (Fig 2), Figure 1 below) and protein level (PMID 24870543 (Fig 3A), Figure 2 below). In addition, a comparative study of Shen C. *et al.*, investigating GAPDH expression within and between a range of cancer types in TCGA, concludes that “GAPDH is not suitable as an internal reference gene for most cancer research, whether RNA or protein analyses” (PMID 37033358 (Fig 1A), Figure 3 below).

[Editorial note: this figure was redacted due to third-party rights. It can be found in Barber RD *et al.*, GAPDH as a housekeeping gene: analysis of GAPDH mRNA expression in a panel of 72 human tissues, *Physiological Genomics* 21, 389–395 (2005). <https://doi.org/10.1152/physiolgenomics.00025.2005>. Figure 2]

[Editorial note: this figure was redacted due to third-party rights. It can be found in Wilhelm et al, Mass-spectrometry-based draft of the human proteome, *Nature* 509, 582–587 (2014). <https://doi.org/10.1038/nature13319>. Figure 3a]

[Editorial note: this figure was redacted due to third-party rights. It can be found in Shen C, et al. Research on the oncogenic role of the house-keeping gene GAPDH in human tumors. *Transl Cancer Res.* 2023. <https://doi.org/10.21037/tcr-22-1972>. Figure 1A].

Lastly, and even more relevant, experiments at the single-cell level have revealed that transcript levels of GAPDH differ considerably among individual cells (PMID 31531674 (Fig 3F), Figure 4 below), ranging over ten orders of magnitude in human and mice samples. Meanwhile, it is conceivable that post-transcriptional regulations could lead to even greater distinctions between single-cell proteomes. Moreover, based on the single-cell analysis of organoids, it has been demonstrated that GAPDH is heterogeneously expressed in tumors, where it showed co-expression with other metabolically important genes (PMID 34105295 (Fig 4B and C), Figures 5 & 6 below) and it was associated with an oxidative stress response (PMID 34105295 (Fig 5), not shown here), similar to the findings in our work.

[Editorial note: this figure was redacted due to third-party rights. It can be found in Taken from Lin et al, Evaluating stably expressed genes in single cells. *Gigascience* 8(9):giz106 (2019). <https://doi.org/10.1093/gigascience/giz106>. Figure 3F]

[Editorial note: this figure was redacted due to third-party rights. It can be found in Taken from Zhao Y et al., Single-Cell Transcriptome Analysis Uncovers Intratumoral Heterogeneity and Underlying Mechanisms for Drug Resistance in Hepatobiliary Tumor Organoids. *Advanced science (Weinheim)* 8(11), e2003897 (2021). <https://doi.org/10.1002/adv.202003897>. Figure 4B]

[Editorial note: this figure was redacted due to third-party rights. It can be found in Zhao Y et al., Single-Cell Transcriptome Analysis Uncovers Intratumoral Heterogeneity and Underlying Mechanisms for Drug Resistance in Hepatobiliary Tumor Organoids. *Advanced science (Weinheim)* 8(11), e2003897 (2021). <https://doi.org/10.1002/adv.202003897>. Fig 4C].

Overseeing this, we believe that a FACS experiment for GAPDH will not bring new insight, and that the concordant difference in expression of proteins acting in oxidative stress response provides sufficient support to conclude that cells differ in metabolic state.

Nevertheless, to address potential concerns about the differences in GAPDH expression levels, we extended our manuscript by a sentence (line 386), summarizing our argumentation and giving references to the previously described diversity of GAPDH among tissues and individual cells:

“Since these proteins are e.g. involved in oxidative stress response (GAPDH, CAT, ARG1) this observation indicates that cells with different metabolic activity co-exist in the initial cell population, yet such diversity does not dictate the responsiveness to IFN- γ . Moreover, these data show that GAPDH differed 8-fold in expression between individual cells (Fig. 6E), calling into question its conventional status as a housekeeping protein. Indeed, it was found before that GAPDH drastically differs in expression between cell types, both on protein and transcript level (PMID 24870543 and 15769908), and has even been shown to be heterogeneously expressed among individual cells using single-cell RNA analysis (PMID 31531674 and 34105295).”

Reviewer #3:

The additional analysis and clear discussions of the newly added results are well appreciated. I can now recommend the manuscript for publication. The authors addressed all my main comments and concerns:

The authors performed a deeper comparative analysis of DIA-NN and Spectronaut, resulting in a better understanding of the differences between the two search engines' MBR approaches. Moreover, the authors now provide new and improved guidelines for configuring both search engines for DIA-ME, enhancing the value of the manuscript. They also performed the suggested evaluation of false transfer rates (FTR) with single cells, confirming the applicability of DIA-ME, albeit with slightly increased FTR. The FTR results are now appropriately shown at the precursor level instead of the protein level. All relevant code has been uploaded to GitHub, SDRF metadata files have been created, and the raw mass spectrometry data has been uploaded to PRIDE, greatly improving the reproducibility and value of the work for the community. Furthermore, the introduction was revised to position the manuscript as a formalization and evaluation of the technique.

We appreciate the reviewer's recommendation for publication and thank him/her for their remarks that have helped us to significantly refine our work.